# Single-cell transcriptome profiling of the stepwise progression of head and neck cancer

Ji-Hye Choi[1,2,6], Bok-Soon Lee[3,6], Jeon Yeob Jang[2,3,6], Yun Sang Lee[3], Hyo Jeong Kim[3,4], Jin Roh[5], Yoo Seob Shin [3], Hyun Goo Woo [1,2] ✉ & Chul-Ho Kim[3,5] ✉

Head and neck squamous cell carcinoma (HNSCC) undergoes stepwise progression from normal tissues to precancerous leukoplakia, primary HNSCC, and metastasized tumors. To delineate the heterogeneity of tumor cells and their interactions during the progression of HNSCC, we employ single-cell RNA-seq profiling for normal to metastasized tumors. We can identify the carcinoma in situ cells in leukoplakia lesions that are not detected by pathological examination. In addition, we identify the cell type subsets of the Galectin 7B (*LGALS7B*)-expressing malignant cells and *CXCL8*-expressing fibroblasts, demonstrating that their abundance in tumor tissue is associated with unfavorable prognostic outcomes. We also demonstrate the interdependent ligand-receptor interaction of *COL1A1* and *CD44* between fibroblasts and malignant cells, facilitating HNSCC progression. Furthermore, we report that the regulatory T cells in leukoplakia and HNSCC tissues express *LAIR2*, providing a favorable environment for tumor growth. Taken together, our results update the pathobiological insights into cell-cell interactions during the stepwise progression of HNSCCs.

Head and neck squamous cell carcinomas (HNSCCs) are the 6th most common cancer type, including oral, pharyngeal, and laryngeal cancers. The diagnosis and treatment of HNSCC have steadily improved; however, the 5-year survival rates for advanced HNSCCs remain unfavorable at approximately 50–60%[1]. Previously, large-scale genomic profiling studies, such as the Cancer Genome Atlas (TCGA), have demonstrated the molecular heterogeneity of HNSCC, providing new insights into the pathobiology of HNSCC progression[2]. However, studies on bulk tissues are significantly limited in distinguishing the effects of tumor microenvironmental cells, such as immune cells and stromal cells, on cancer progression. Recent advances in single-cell RNA-seq (scRNA-seq) technology have made it possible to overcome these limitations, distinguish changes in gene expression at the single-

cell level, and provide new insights into the interactions among the diverse cell types in tumor tissues[3].

HNSCC undergoes a stepwise progression from normal tissue (NL) to precancerous leukoplakia (LP), followed by primary cancer (CA) and, ultimately, metastatic tumors in the lymph nodes (LN). LP progresses to malignancies at varying rates;[4] therefore, efforts have been made to predict the malignant progression of precancerous lesions based on LP progression rates. However, the molecular mechanisms underlying malignant conversion of LP remain unclear.

In addition, various etiological factors are involved in HNSCC development, including exposure to alcohol or tobacco and human papillomavirus (HPV) infection[5–7]. In particular, HPV infection plays a critical role in HNSCC progression, showing more favorable

[1]Department of Physiology, Ajou University School of Medicine, Suwon, Republic of Korea. [2]Department of Biomedical Science, Graduate School, Ajou University, Suwon, Republic of Korea. [3]Department of Otolaryngology, Ajou University School of Medicine, Suwon, Republic of Korea. [4]Department of Molecular Science and Technology, Ajou University, Suwon, Republic of Korea. [5]Department of Pathology, Ajou University School of Medicine, Suwon, Republic of Korea. [6]These authors contributed equally: Ji-Hye Choi, Bok-Soon Lee, Jeon Yeob Jang. ✉e-mail: hg@ajou.ac.kr; ostium@ajou.ac.kr

prognostic outcomes in HPV-positive patients than in HPV-negative patients[8]. Genomic analyses have also demonstrated that the innate and acquired antiviral immune responses are suppressed in HPV-positive patients[2].

In this study, we perform scRNA-seq profiling of non-tumoral surrounding NL, LP, CA, and LN tissue to delineate the single-cell level alterations and cell-cell interactions that contribute to the stepwise progression of HNSCC from non-neoplastic lesions to metastatic tumors. Considering the mechanical and clinical impacts of HPV, we carefully analyze the single-cell transcripts according to the HPV infection status. We demonstrate the stepwise alterations in cell composition during HNSCC progression, identifying cell clusters for malignant, stromal, and immune cells. Our results reveal the cell-cell interactions between tumor cells and their microenvironment, providing key mechanistic and clinical insights into HNSCCs.

## Results

### scRNA-seq profiling of the stepwise progression of HNSCC

scRNA-seq profiles were obtained from 23 patients of HNSCC, including the tissue types of NL ($n = 9$), LP ($n = 4$), CA ($n = 20$), and LN ($n = 4$) (Fig. 1a and Supplementary Table 1). A total of 54,239 cells were clustered into 16 clusters, which were designated into 9 cell types based on the differential expression of known cell type marker genes, including epithelial cells, fibroblasts, endothelial cells, myocytes, immune NK/T cells, B/plasma cells, macrophages, dendritic cells, and mast cells (Fig. 1b and Supplementary Fig. 1a). The cell clusters were related with tissue type, HPV infection status, and patient samples (Fig. 1c). LP tissues from the HPV-positive patients were unavailable; therefore, they were excluded from the analysis. When the cell composition was evaluated according to HPV infection status, HPV-positive tumors exhibited a lower proportion of fibroblasts (11.49% vs. 1.02%)

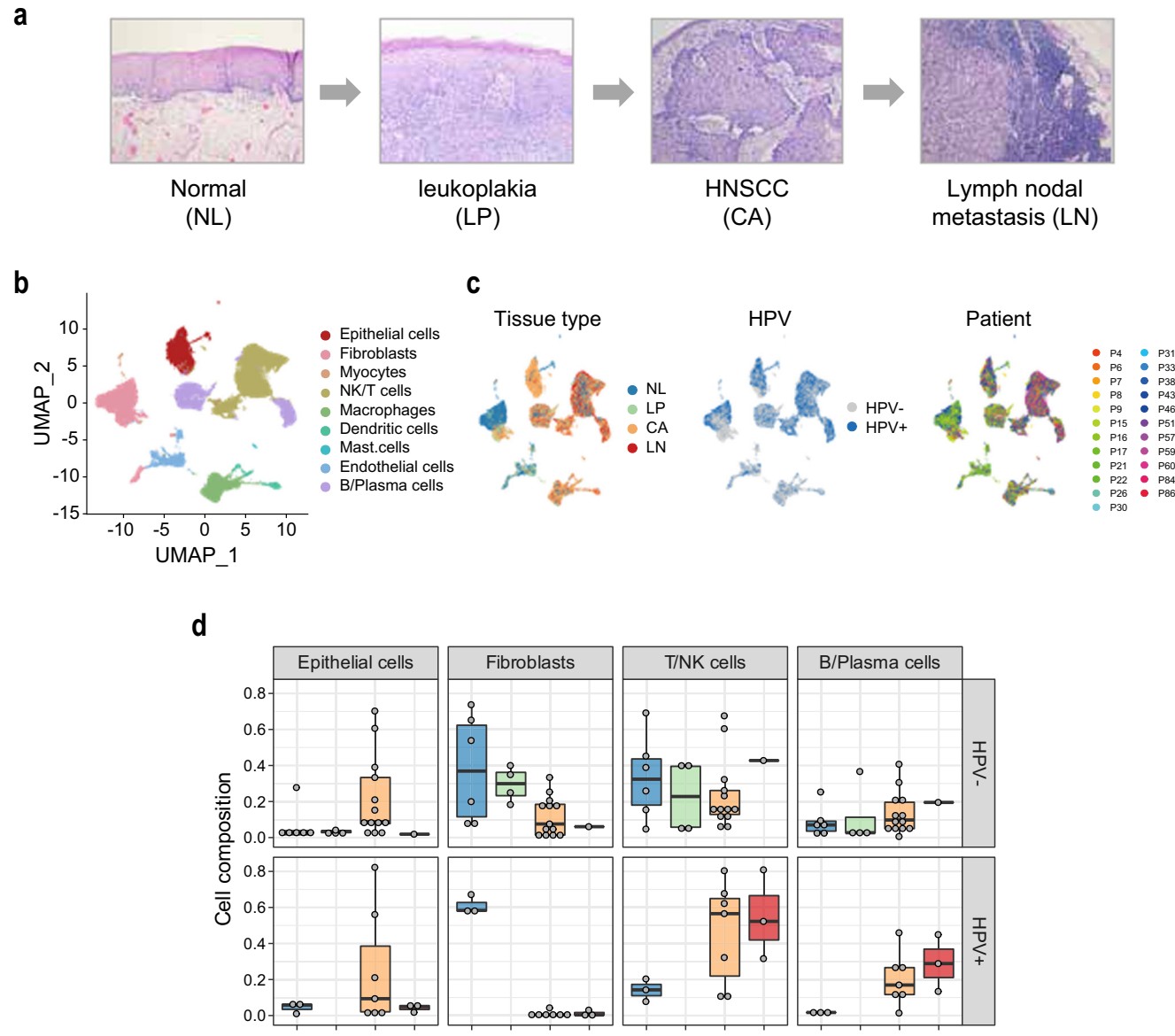

**Fig. 1 | scRNA-seq profiling of the stepwise progression of HNSCC. a** A schematic diagram of the stepwise progression of HNSCC. **b** Cell types of the 54,239 cells are indicated in a UMAP plot. **c** Distribution of tissue types, HPV infection status, and individual patients are shown. **d** Cell compositions of the major cell types of epithelial cells ($n = 6106$), fibroblasts ($n = 12,336$), NK/T cells ($n = 17,869$), and B/Plasma cells ($n = 7,437$) are shown according to the tissue types and patients' HPV infection status. Data points represent the average value of the cell type proportions in each sample. Box plots show the median (center line), the upper and lower quantiles (box), and the range of the data (whiskers). Source data are provided in a Source Data file.

but higher proportions of NK/T cells (48.50% vs. 25.05%) and B/plasma cells (22.79% vs. 13.87%) compared to those in HPV-negative tumors (Fig. 1d and Supplementary Fig. 1b). This finding was consistent with the previous studies, indicating functional alterations of the immune cells in HPV-positive tumors[9, 10].

## DNA copy number-dependent deregulation of *TP63* and *ATP1B3* in LP tissues

Next, we inferred the recurrent DNA copy number aberrations (CNAs) using the inferCNV method[11] with an improved modification excluding the effects of the co-expressed functional gene clusters and non-recurrent aberrations (for details, see "Methods"). The epithelial cells of the CA and LN tissues exhibited profound CNAs but did not the other stromal cells (Supplementary Fig. 2a). Consistent with the previous studies[12, 13], HPV-negative tumors showed higher DNA copy gains than HPV-positive tumors (average number of genes with CNAs per cell = 245.2 vs. 111.7, $P < 3.0 \times 10^{-16}$, Fig. 2a). In LP tissues, we observed the 80 epithelial cells harboring CNAs similar to malignant cells, which we designated as carcinoma in situ (CIS) cells. Comparing the CNAs between NL and LP tissues in HPV-negative patients, CIS cells showed prominent DNA copy gains at 1q, 3q, 8q, 20p, and 22q, and losses at 3p, 10p, and 10q, confirming the previous findings[14] (Fig. 2b). The CIS cells begun to express TNF-α/NF-kB, apoptosis, P53, estrogen response, hypoxia, PI3K/AKT/mTOR, and glycolysis-related genes, and which were further aggravated in the malignant cells (Fig. 2c). Gene-level analysis revealed that the CIS, compared to the epithelial cells, expressed the known tumor marker genes [*e.g.*, *CXCL1*, *EFNA1*, *TM4SF1*, *ELF3*, and keratin cytoskeletal genes (*i.e.*, *KRT19*, *KRT13*, *KRT18*, and *KRT8*)], indicating their malignant characteristics[15–17] (fold difference > 1, permuted Student's t-test $P < 0.0001$, Fig. 2d).

Previously, CNA-dependent transcriptional dysregulation has been suggested to play a driving role in cancer progression[18]. With this concern, we identified the CNA-dependent genes which had differentially altered DNA copy numbers with concomitant transcriptional deregulation across the tissue types ($n = 8$, permuted Student's t-test $P < 0.001$ and fold difference > 1, Supplementary Fig. 2b, see Supplementary Methods). *TP63* and *ATP1B3* at chromosome 3q exhibited the most prominent CNA-dependent transcriptional deregulation in the HPV-negative CIS cells (Fig. 2e). We could validate this result in the HPV-negative TCGA-HNSCC data, showing significant correlations between CNAs and transcription levels of *TP63* ($r = 0.19$, $P = 2.5 \times 10^{-3}$) and *ATP1B3* ($r = 0.37$, $P = 2.0 \times 10^{-9}$) (Fig. 2f). Oncogenic activities of these genes have been shown in various cancer types[19, 20], which could be validated by performing siRNA-mediated knockdown experiments. Treatment with siRNAs targeting *TP63* (si*TP63*) or *ATP1B3* (si*ATP1B3*) significantly suppressed tumor-promoting functions of HNSCC cells, including cell viability, tumor sphere formation, migration, and invasion (Fig. 2g, h, I and Supplementary Fig. 3). These results consistently support that the CNA-dependent expression of *TP63* and *ATP1B3* in premalignant LP lesions plays a critical role in the progression of HNSCC.

## Malignant cell cluster CC1 had an aggressive phenotype

Next, we assigned the epithelial cells in CA and LN tissues as malignant cells ($n = 5,113$). To assess the transcriptional heterogeneity, the malignant cells were sub-classified into six clusters (CC0 to CC5, Fig. 3a, *top*, see Supplementary Methods). We observed that each pair-matched primary and metastatic tumor (CA and LN) was clustered together as demonstrated previously[21] (Fig. 3a, *bottom*). In addition, we observed that the cell composition of the malignant cell clusters of the metastasized tumors was very similar to that of the pair-matched primary tumors, revealing that each pair of the primary and metastatic tumors was clustered together in the hierarchical cluster analysis (Fig. 3b). This result suggests that the transcriptional diversity of the primary tumors is conserved in metastatic tumors.

Next, we performed deconvolution analyses to evaluate the cell type proportions in bulk RNA-seq data of TCGA-HNSCC ($n = 500$, Supplementary Methods). Unlike scRNA-seq data, deconvolution analysis revealed that the malignant cells were the most abundant cell type (79.17%) followed by fibroblasts (10.07%) (Supplementary Fig. 4a). Among the malignant cells, CC0 and CC1 were the most abundant malignant cell clusters (Fig. 3c). Similar results on malignant cell composition were observed in independent datasets of GSE41613 ($n = 97$)[22], GSE42743 ($n = 74$)[22], and GSE65858 ($n = 270$)[23], demonstrating the robustness of our result (Supplementary Fig. 4b). Interestingly, we found that the malignant cell clusters were closely located according to the HPV infection status (Fig. 3d, *top*). CC0 and CC4 cells were mostly HPV-positive tumors, whereas the other clusters were HPV-negative tumors (Fig. 3d, *bottom*). This finding indicates that our malignant cell clusters harbor distinct transcriptomic programs related to HPV infection. In addition, the tumor subsites (*e.g.*, oropharynx and oral cavity) have been shown to associate with HPV infection status;[24] however, we could not observe significant associations between the malignant cell clusters and tumor subsites (Fig. 3e).

We also evaluated whether our malignant cell clusters could be found in independent scRNA-seq datasets. Subtyping of the malignant cell using nearest template prediction (NTP) analysis could re-identify our malignant cell clusters in GSE103322 (77.16%) and GSE164690 (99.10%), respectively (Supplementary Fig. 5); however, our cell subtypes did not cover all the heterogeneity of the malignant cells, which might be due to a small number of patients being sampled. This may support that our malignant cell clusters represent most of the malignant cell types in HNSCC. In addition, we examined the expression of the previous single-cell malignant cell programs of HNSCC in each of the malignant cell clusters[21]. The CC0 expressed hypoxia and epithelial differentiation-related genes while the CC1 expressed the partial epithelial-mesenchymal transition (p-EMT)-related genes (Fig. 3f). The p-EMT has been suggested to localize in the leading edge in proximity to cancer-associated fibroblasts (CAFs) and are promoted through paracrine interactions between CAFs and malignant cells, resulting in aggressive progression of HNSCC. Supporting this, we could demonstrate that the CC1-high group (>70 percentile of the CC1), compared to the CC1-low group, had more unfavorable clinical outcomes of overall survival (OS; hazard ratio [HR] = 1.38, $P = 0.02$) and recurrence-free survival (RFS; HR = 1.69, $P = 0.006$) in TCGA-HNSCC, although the other malignant cell clusters did not (Fig. 3g, *top*). This finding could be validated using the independent datasets of GSE41613 (OS, HR = 2.30, $P = 0.003$) and GSE42743 (OS, HR = 2.16, $P = 0.016$) (Fig. 3g, *bottom*). Multivariate analyses also revealed the significance of the associations of CC1 with worse prognostic outcomes of HNSCC (GSE41613, OS, HR = 2.68, $P = 0.001$; GSE42743, OS, HR = 2.33, $P = 0.011$, Supplementary Table 2). Taken together, we suggest that the abundance of CC1 is associated with the expression of HPV-negative tumor-like features acquiring an aggressive phenotype.

Next, we identified the differentially expressed genes that may represent each cluster (Wilcoxon Rank Sum test $P < 0.001$, fold difference > 1, Supplementary Fig. 6). We found that the CC1 prominently expressed *LGALS7B* (Fig. 3h), which could be validated by showing the expression of *LGALS7B* in the CC1-high samples (*i.e.*, C04, C30, C51, C26, C07, C06, C31, and C15) but not in the CC1-low samples (C38 and C09) (Fig. 3i and Supplementary Fig. 7). Galectin-7 has a tumor-promoting function in diverse cancer types, including HNSCC[25–27], therefore, it is plausible that the aggressive feature of CC1 might be associated with the expression of p-EMT and *LGALS7B*.

## Fibroblasts-derived *COL1A1* expression interacts with *CD44* in malignant cells

Fibroblasts are the second most common cell type in the deconvolution analysis (see Supplementary Fig. 4a). We could identify the stepwisely expressed genes during progression from NL to LP and CA

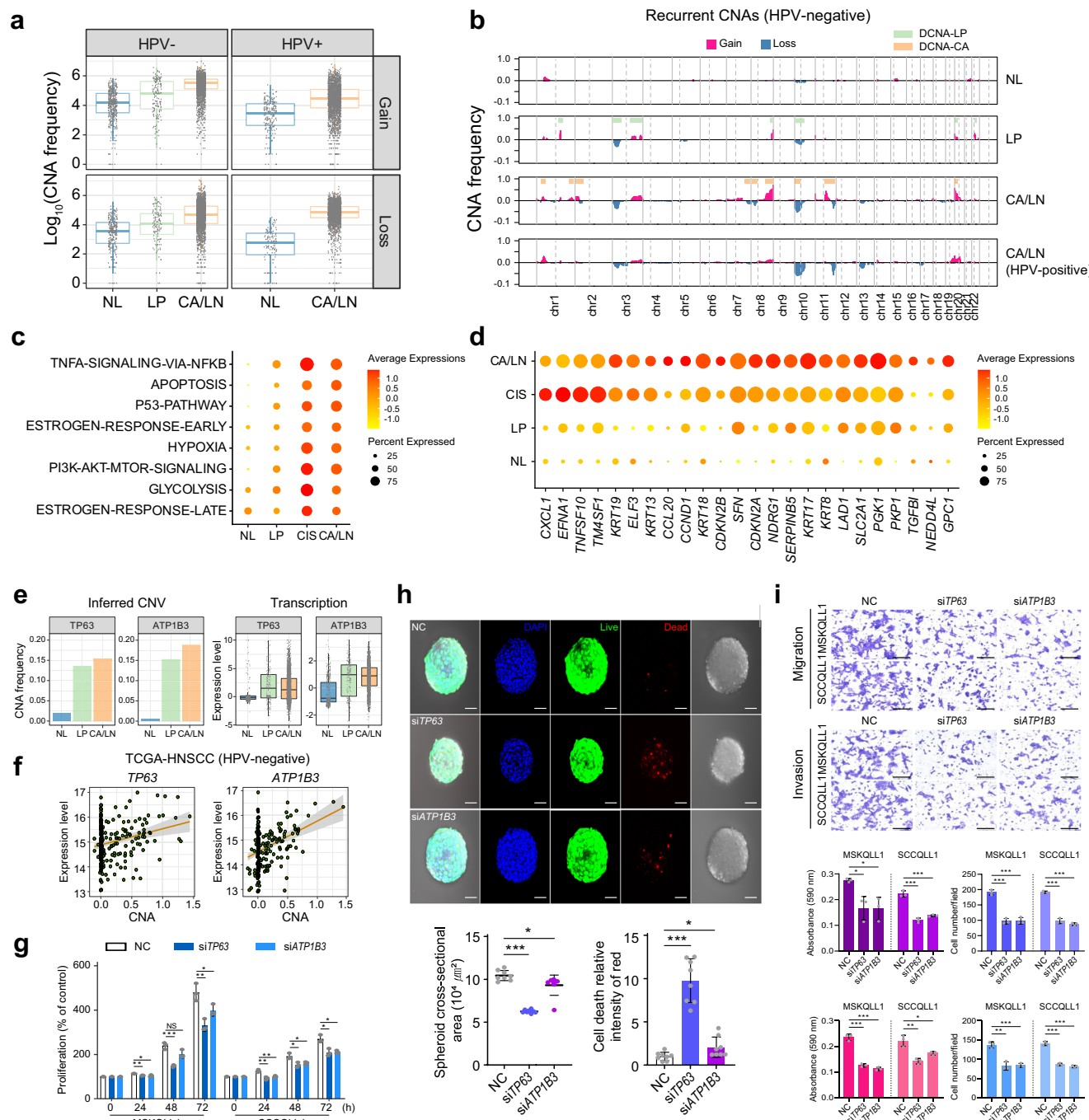

**Fig. 2 | DNA copy number-dependent deregulation of *TP63* and *ATP1B3* promotes HNSCC progression. a** For each tissue type, log-scaled frequencies of the CNA gains (*top*) and losses (*bottom*) in the HPV-negative and HPV-positive samples are shown. **b** CNA frequencies are shown in chromosomal order for each HPV-negative tissue type. **c**, **d** Dot plots show the differential expression of the Hallmark gene sets (**c**) and genes (**d**) in the epithelial cells across the tissue types of NL, LP, CIS, and CA/LN tissues. **e** Frequencies of the CNAs (*left*) and the expression levels (*right*) of *TP63* and *ATP1B3* genes in HPV-negative patients are shown. **f** DNA copy number-correlated transcriptional expression of *TP63* and *ATP1B3* are shown in the HPV-negative TCGA-HNSCC patients (*n* = 240). **g** The MSKQLL1 and SCCQLL1 cells are treated with the non-target control (NC), si*TP63*, or si*ATP1B3*, and their viabilities are shown at indicated times. **h** FaDu cells are transfected with NC, si*TP63*, or si*ATP1B3* and cultured in spheroids using hanging drop plates, and the cell viability is shown (*top*). The cross-section area of FaDu cells (*bottom left*) and the numbers of dead cells (*bottom right*) with the treatment of NC, si*TP63*, or si*ATP1B3*, are shown. *n* = 8 biologically independent experiments. Scale bar, 50 μm. **i** MSKQLL1 and SCCQLL1 cells are transfected with NC, si*TP63*, or si*ATP1B3*, and their cell proliferation, migration, and invasion are shown. *n* = 3 biologically independent experiments. Scale bar, 100 μm. In **g**–**i** data are shown as mean ± SD. *,*P* < 0.05; **,*P* < 0.01; ***,*P* < 0.001; ns, nonsignificant by the Student two-tailed unpaired *t*-test. In **a**, **e**, box plots show median (center line), the upper and lower quantiles (box), and the range of the data (whiskers). Source data are provided in a Source Data file.

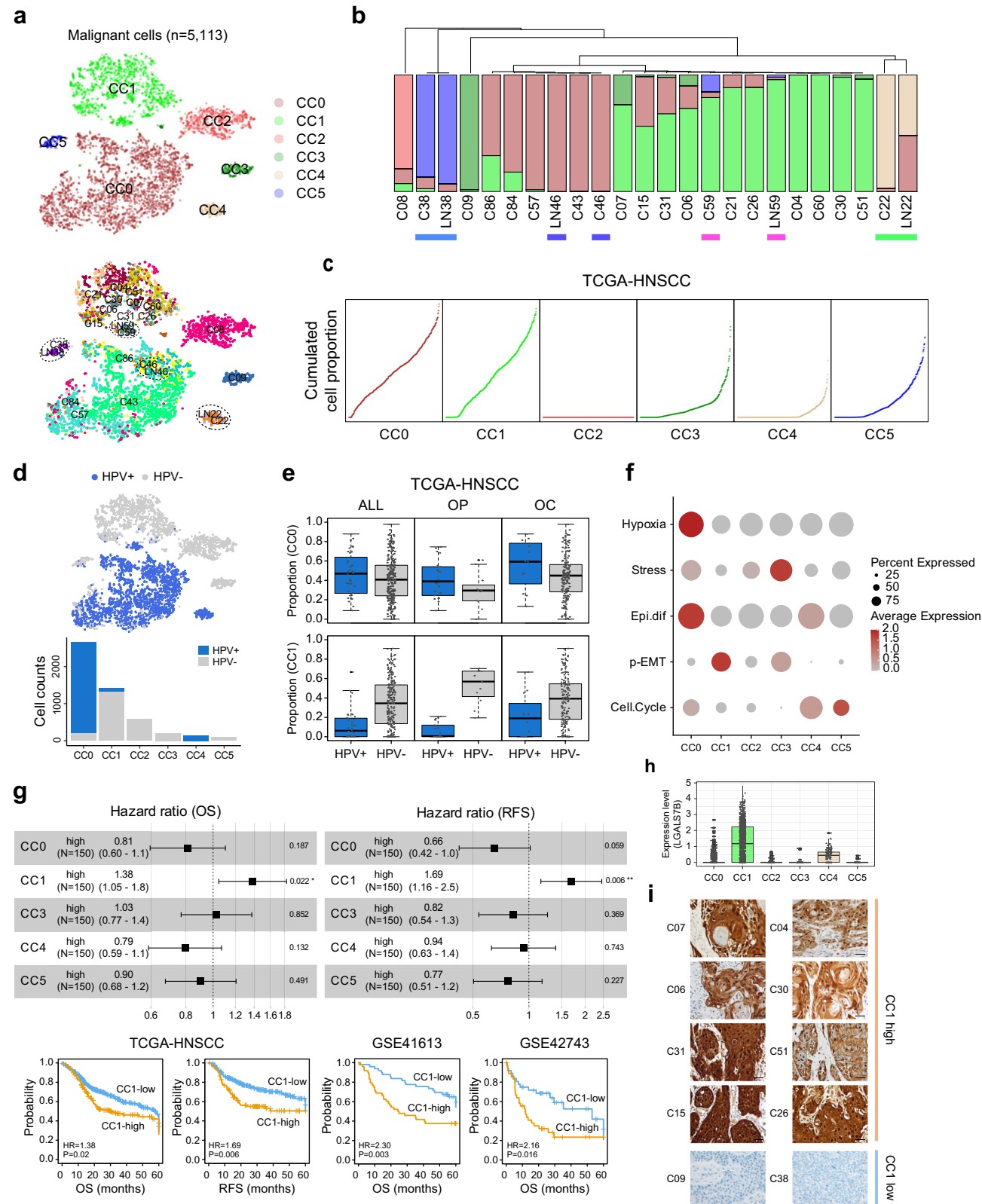

tissue in each cell type of epithelial/malignant cells ($n = 17$, e.g., KRT6A) and fibroblasts ($n = 41$, e.g., POSTN) (Supplementary Fig. 8), and which could be demonstrated more clearly by a pseudo-time trajectory analysis[28] (Fig. 4a). Interestingly, we observed that the pseudo-time of LP was closer to that of CA or LN rather than that of NL, which may indicate that the premalignant LP fibroblasts already acquired a CAF-like feature (Fig. 4b). To further delineate the effects of the

transcriptional alteration of fibroblasts on cancer progression, we analyzed the ligand-receptor (LR) interaction between fibroblasts and malignant cells (for details, see Supplementary Methods). We aimed to identify interdependent LR pairs between malignant cells and fibroblasts, which exhibit receptor expression in malignant cells, but whose corresponding ligands are expressed only in fibroblasts and not in malignant cells. We could identify the seven LR pairs showing

**Fig. 3 | Malignant cell cluster with LGALS7 expression shows an aggressive phenotype. a** Malignant cell clusters (CC0-CC5, *top*) and samples (*bottom*) are shown in t-SNE plots. Paired primary and metastatic tumors are indicated (*dotted circle*). **b** The hierarchical clustering of the malignant clusters shows a similar cell proportion of the paired primary and metastatic tumors (*underlined*). **c** The cumulative cell proportion of the malignant cell clusters is shown in TCGA-HNSCC ($n = 500$). **d** The HPV infection status is indicated in a t-SNE plot (*top*). Cell counts of the HPV-positive and HPV-negative cells are shown across the malignant cell clusters (*bottom*). **e** According to the tumor subsites for all the patients (ALL, $n = 500$), OP ($n = 71$), and OC ($n = 308$), the proportions of the CC0 and CC1 malignant cell clusters in the HPV-positive and HPV-negative tissues are shown. **f** A dot plot shows the average expression levels of the single-cell malignant programs of HNSCC, including hypoxia, stress, epithelial differentiation, p-EMT, and cell cycle in each malignant cluster. **g** Forest plots show the hazard ratios for OS and RFS between the high- and low-proportion groups (>70th percentile) for each malignant cell cluster in the deconvoluted TCGA-HNSCC data (*top*). CC2 is not shown because its 70th percentile of cell proportion is zero. Hazard ratios and 95% confidence intervals (CI) are indicated. *,$P < 0.05$ and **,$P < 0.01$, log-rank test. Kaplan–Meier's plot analyses for OS (*left*) or RFS (*right*) between the CC1-high and CC1-low groups are shown in TCGA-HNSCC, GSE41613, and GSE42743, respectively (*bottom*). $P = 0.02$ (OS; TCGA-HNSCC), $P = 0.006$ (RFS; TCGA-HNSCC), $P = 0.003$ (OS; GSE41613), $P = 0.016$ (OS; GSE42743), log-rank test. Follow-up time for OS and RFS is truncated to 5 years. **h** Expression levels of *LGALS7B* in the malignant cell clusters are shown. **i** Representative images of immunohistochemical staining for LGALS7B expression are shown in the CC1-high and CC1-low samples. $n = 1$. Scale bar, 100 μm. In **e**, **h**, box plots show the median (center line), the upper and lower quantiles (box), and the range of the data (whiskers). Source data are provided in a Source Data file.

interdependent and stepwise expression of the ligands in fibroblasts (*i.e.*, *COL1A1, COL1A2, COL6A3, THBS1, THBS2, TNC,* and *LAMA4*) and their corresponding putative receptors in malignant cells (*i.e.*, *CD44, ITGB1, SDC4, CD47,* and *ITGA6*, Fig. 4c). Remarkably, the *COL1A1-CD44* ligand-receptor pair showed the most prominent and interdependent expression (Fig. 4d, *top*), and which could be validated by Puram's scRNA-seq data (Fig. 4d, *bottom*). Moreover, *COL1A1* expression was most significantly correlated with the pseudo-time of the fibroblasts ($r = 0.57$, $P < 2.2 \times 10^{-16}$, Supplementary Fig. 9a). We could successfully demonstrate the correlated expression between *COL1A1* in fibroblasts and *CD44* in epithelial cells ($r = 0.43$, $P = 7.47 \times 10^{-3}$, Fig. 4e, and Supplementary Fig. 9b). However, *CD44* expression in fibroblasts was not correlated with *COL1A1* expression (Supplementary Fig. 9c). These results suggest that the *CD44-COL1A1* interaction occurs between fibroblasts and malignant cells but not in an autocrine manner. Immunohistochemical analysis also demonstrated the prominent COL1A1 expression in fibroblasts near the CD44-expressing malignant cells (Fig. 4f and Supplementary Fig. 9d).

In addition, we evaluated the functions of *COL1A1-CD44* interaction by performing siRNA-mediated knockdown experiments for *COL1A1* (si*COL1A1*) and *CD44* (si*CD44*). Two HNSCC cell lines expressing *CD44* (MSKQLL1 and SCCQLL1, Supplementary Fig. 10a) and the CAFs expressing *COL1A1* (CAF48) were selected (Supplementary Fig. 10b). We observed that the treatment with collagen (type I) induced tumor cell invasion, and which was abolished by the transfection with si*CD44* (Fig. 4g). In addition, the co-culture of MSKQLL1 and CAF48 enhanced tumor cell migration but did not the co-culture of the si*CD44*-transfected MSKQLL1 cells with CAF48 or the MSKQLL1 cells with si*COL1A1*-transfected CAF48 cells (Fig. 4h). Taken together, we suggest that *COL1A1* in fibroblasts and *CD44* epithelial/ malignant cells interact interdependently, facilitating tumor progression.

### *CXCL8*-expressing CAFs aggravate HNSCC progression

The CAFs from tumor tissues (CA and LN) were further stratified into five clusters (CF0 to CF4), revealing their representative marker genes (Fig. 5a and Supplementary Fig. 11a). When the CAF clusters were compared with the previous CAFs in Puram's study[21], CF0 and CF4 were similar to the Puram's subtype CAF1 expressing its marker genes (*e.g.*, *CTHRC1, COL1A1, COL3A1, POSTN,* and *MFAP2*, Fig. 5b). However, CF4 was different from CF0, showing higher proliferation activity (Supplementary Fig. 11b). CF2 and CF3 were similar to Puram's CAF2 and myofibroblasts, expressing their marker genes such as *CXCL12* and *NDUFA4L2*. Interestingly, we identified a CAF subtype, CF1, expressing *CXCL8* (IL-8 coding gene). CF1 showed a distinct expression pattern differing from the other CAF clusters in the two-dimensional diffusion map embedding analysis[29] (Fig. 5c, *left*). Pseudo-time projection analysis also revealed that CF1 was the last trajectory, implying that CF1 was the most progressed fibroblast (Fig. 5c, *right*). Moreover, we demonstrated that the abundance of CF1 was significantly correlated

with the *CXCL8* expression levels in independent datasets of TCGA-HNSCC ($r = 0.38$, $P = 2.72 \times 10^{-18}$), GSE41613 ($r = 0.74$, $P = 3.41 \times 10^{-18}$), GSE42743 ($r = 0.62$, $P = 3.54 \times 10^{-9}$), and GSE65858 ($r = 0.68$, $P = 9.58 \times 10^{-38}$, Fig. 5d). When we compared the proportion of the CAF subtypes according to the HPV infection status, most CF1 cells were observed in HPV-negative tumors, but not in HPV-positive tumors ($P = 1.33 \times 10^{-7}$, Supplementary Fig. 11c, *left*). TCGA data also showed that CF1 cells were more abundant in the HPV-negative tumors (10.15%) than HPV-positive tumors (5.04%) ($P = 5.26 \times 10^{-7}$, Supplementary Fig. 11c, *right*).

Next, we assessed whether the abundances of CF1 and CC1 are associated with each other because both CF1 and CC1 abundances had aggressive features. As expected, CC1 and CF1 abundances were significantly correlated in the pooled HNSCC data ($r = 0.41$, $P = 1.11 \times 10^{-38}$, Fig. 5e), implying that CF1 and CC1 may interact together. Indeed, galectin-7 (expressed in CC1) promotes metastasis in various cancer types, including HNSCC[30]. IL-8 (expressed in CF1) is produced by multiple cell types promoting tumor progression[31, 32]. To evaluate this hypothesis, we demonstrated that galectin-7 treatment significantly induced *CXCL8* expression in various CAF cells (CAF30, CAF57, CAF58, and CAF70) as well as HNSCC cells (SCCQLL1 and SNU1066) (Fig. 5f). By contrast, IL-8 treatment did not induce *galectin-7* in HNSCC cells. (Supplementary Fig. 12). These results suggest that galectin-7 activates both malignant cells (CC1) and fibroblasts (CF1) to express IL-8, facilitating the aggressive progression of HNSCC.

### LAIR2 expression in Treg cells is associated with HNSCC progression

Among the immune cells, NK/T cells were the most common cell type in the scRNA-seq data. The cells with chromosomal aberrations were re-assigned as the non-determined cell type and excluded in the following analyses (see "Methods"). Then, the NK/T cell cluster was further stratified into six subclusters; naïve T cells, CD8+CCL5+, CD4+FOXP3+(regulatory T cell, Treg), CD4+CD154+ (follicular helper T cell), cycling T cells, and NK cells (Fig. 6a, *top*). The T cell cluster showing higher proliferation activity was designated as cycling T cells, showing higher expression of CD8 coding genes (*i.e.*, *CD8A* and *CD8B*) than *CD4* as previously described[33] (Fig. 6a, *bottom*, and Supplementary Fig. 13a). During HNSCC progression, the proportions of cycling T cells and CD4+ T cells were increased, whereas the proportions of naïve T cells and CD8+ T cells were decreased regardless of their HPV infection status (Fig. 6b). Pseudo-time trajectory analysis demonstrated the stepwise transition of the T cells according to the tissue types (Fig. 6c). Notably, Tregs in LP exhibited two peaks of pseudo-time distribution, indicating the alteration of the T cell states in the premalignant lesions. In particular, we observed the prominent expression of leukocyte-associated immunoglobulin-like receptor 2 (*LAIR2*, *CD306*) in the Tregs in LP (permuted Student's t-test $P = 2.50 \times 10^{-9}$, fold difference = 2.32, Fig. 6d, and Supplementary Fig. 13b). Although the functions of *LAIR2* are largely unknown, *LAIR2*

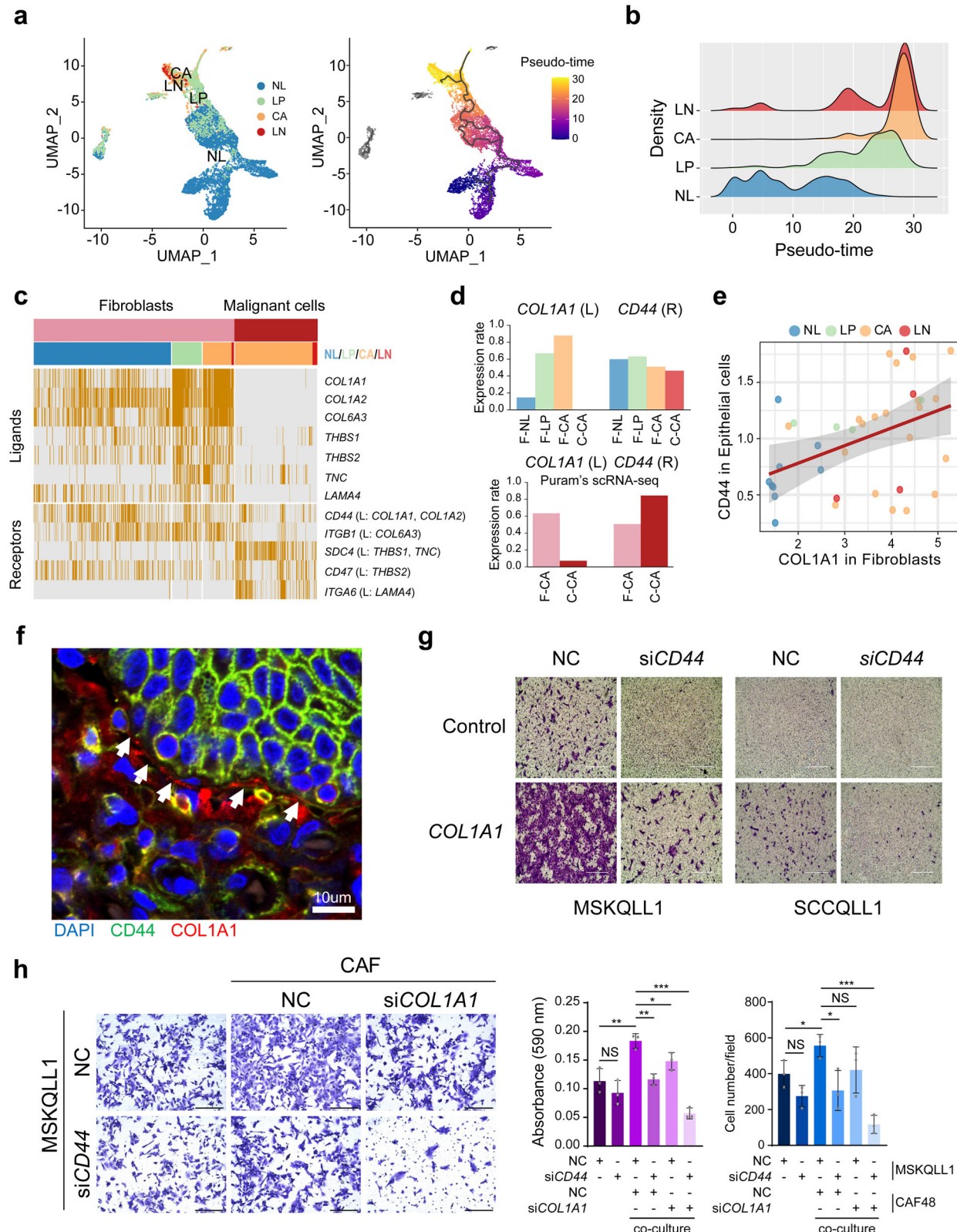

has been known as a soluble collagen receptor that activates pro-inflammatory processes[34]. Therefore, *COL1A1* expression in the fibroblasts may activate the *LAIR2*-mediated downstream signaling in Tregs. In support of this, we observed that *COL1A1* expression levels in fibroblasts were weak but significantly correlated with *LAIR2* expression levels ($r = 0.48$, $P = 2.48 \times 10^{-3}$) and the abundance of Tregs ($r = 0.38$, $P = 2.60 \times 10^{-2}$, Fig. 6e). In addition, we demonstrated that

collagen treatment significantly induced *FOXP3* expression in mouse Tregs (Fig. 6f and Methods). In addition, by performing fluorescence-activated cell sorting (FACS) analysis, we confirmed that collagen treatment increased the number of both FOXP3+CD25+ Tregs and LAIR2+FOXP3+CD25+ Tregs obtained from the PBMCs in the HNSCC patients (Fig. 6g, h, and Supplementary Fig. 13c, and "Methods"). Taken together, we suggest that the collagen from the fibroblasts interacts

**Fig. 4 | Fibroblasts-derived COL1A1 expression interacts with CD44 in malignant cells. a** Tissue types of NL, LP, CA, and LN (*left*) and the pseudo-times (*right*) of the cells are shown in UMAP plots. **b** Distributions of the pseudo-time in the fibroblasts across the tissue types are shown. **c** A heatmap shows the interdependent expression of the ligand-receptor pairs with stepwise expression during HNSCC progression between fibroblasts and malignant cells. **d** The cell proportions for expressed cells of *COL1A1* and *CD44* are shown in fibroblasts (F-) and malignant cells (C-) across the tissue types (*e.g.*, F-NL, fibroblasts in NL; C-NL, malignant cells in NL) from our data (*top*) and that of GSE103322 (*bottom*). **e** The correlation plot shows *COL1A1* expression levels in fibroblasts and *CD44* expression levels in epithelial cells in HNSCC patients. The gray shading represents 95% confidence

interval (CI). **f** An immunohistochemical image shows the expression of collagen and CD44 in fibroblasts (*red*) and neighbored malignant cells (*green*), respectively. *n* = 3. **g** Invasion of the MSKQLL1 and SCCQLL1 cells treated with siRNAs targeting *CD44* (si*CD44*) or non-target control (NC) in the presence or absence of collagen (1 μg/mL). *n* = 3. Scale bar, 400 μm. **h** MSKQLL1 cells and CAF cells are co-cultured with the treatment of NC, si*CD44*, or si*COL1A1*, and the images of the migrated cells (*left*) and the measured cell population (*right*) are shown. *n* = 3 biologically independent experiments. Scale bar, 100 μm. Data are shown as mean ± SD. *,*P* < 0.05; **,*P* < 0.01; ***,*P* < 0.001; ns, nonsignificant by the Student two-tailed unpaired *t*-test. Source data are provided in a Source Data file.

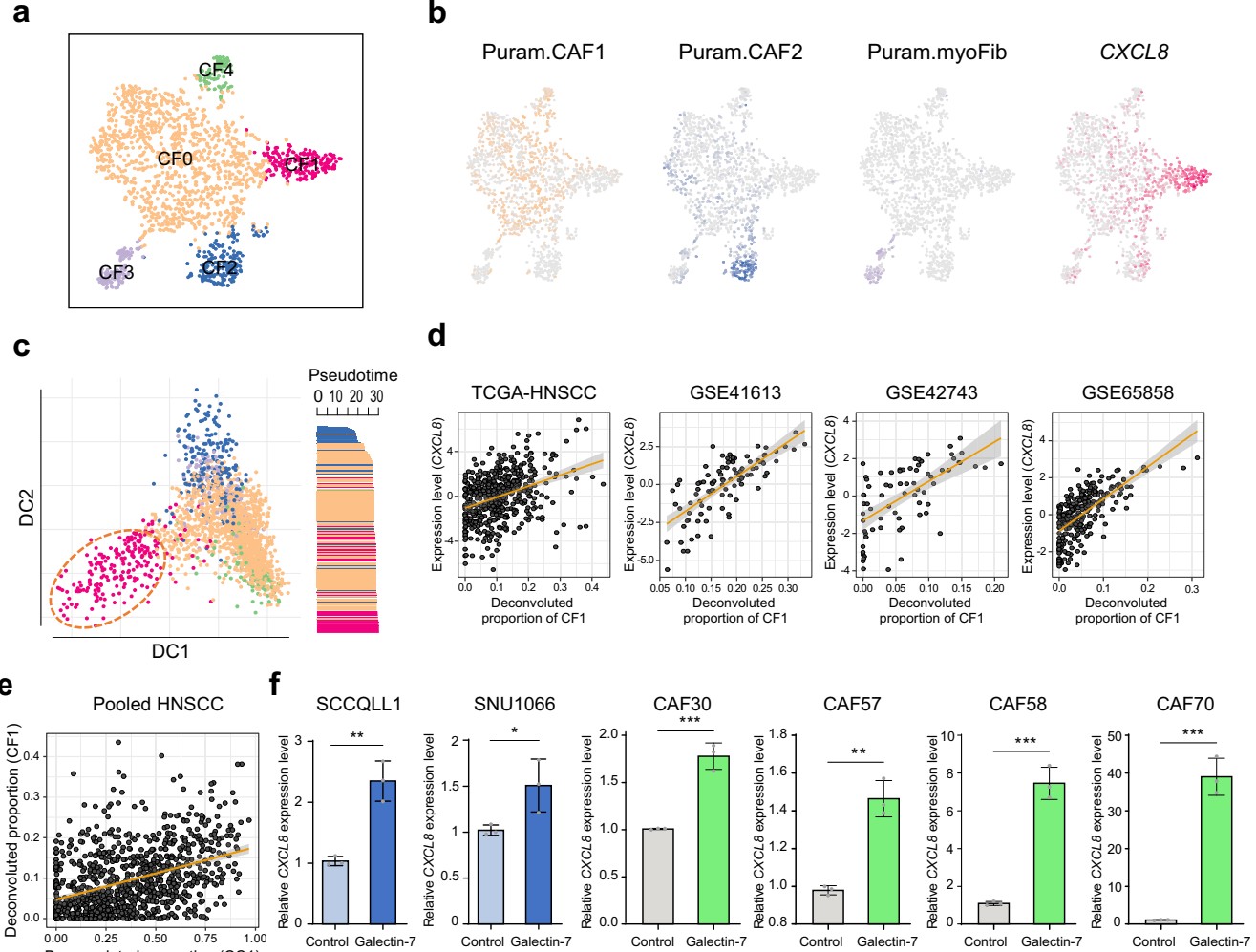

**Fig. 5 | *CXCL8*-expressing CAFs aggravate HNSCC progression. a** CAF clusters are shown in a t-SNE plot (CF0-CF4). **b** Expression levels of the markers for CAF1, CAF2, myofibroblasts (myoFib), and a CF1 marker, *CXCL8*, are shown in t-SNE plots. **c** A diffusion map plot shows the CAF clusters (CF0-CF4, *left*). Pseudo-time scores across CAF clusters are shown (*right*). **d** The correlation plots show *CXCL8* expression levels and the cell proportion of CF1 in independent data sets of TCGA-HNSCC, GSE41613, GSE42743, and GSE65858. **e** A plot shows the correlation between the proportion of CC1 and CF1 in pooled HNSCC data (TCGA-HNSCC,

GSE41613, GSE42743, and GSE65858, n = 941). In **d**, **e** The gray shading represents 95% confidence interval (CI). **f** *CXCL8* expression levels in the presence or absence of *galectin-7* are shown in HNSCC cells (SCCQLL1 and SNU1088) and CAF cells (CAF30, CAF57, CAF58, and CAF70). *P* = 0.0025 (SCCQLL1), *P* = 0.0457 (SNU1088), *P* = 0.0007 (CAF30), *P* = 0.0011 (CAF57), *P* = 0.0002 (CAF58), *P* = 0.0002 (CAF70). *n* = 3 biologically independent experiments. Data are shown as mean ± SD. *,*P* < 0.05; **,*P* < 0.01; ***,*P* < 0.001; ns, nonsignificant by the Student's two-tailed unpaired *t*-test. Source data are provided in a Source Data file.

with Tregs, inducing LAIR2 and FOXP3 expression and providing a favorable microenvironment for tumor progression.

## Discussion

As the next step of TCGA that elucidates molecular heterogeneity using large-scale, multi-omic cancer profiles, a pre-cancer atlas detailing the sequential alterations would be advantageous for

delineating driver events during cancer progression. In this study, we aimed to profile the stepwise progression of HNSCC from NL to pre-cancerous LP, CA, and metastatic LN tissues, and demonstrated the single-cell-level transcriptomic dynamics during the stepwise progression of HNSCC. The key findings regarding the interactions among the malignant cells, fibroblasts, and immune cells are summarized in Fig. 7.

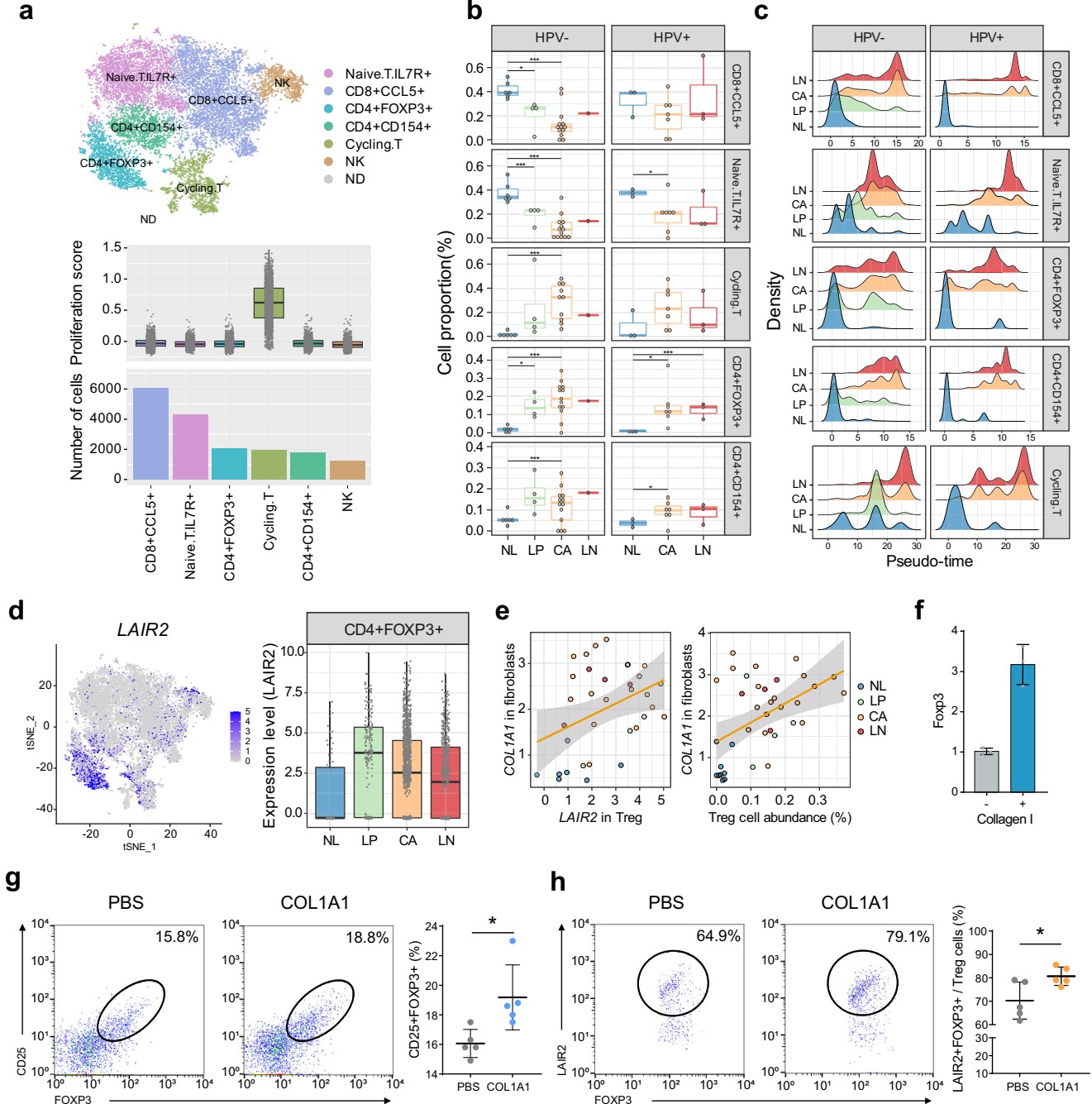

**Fig. 6 | LAIR2 expression in CD4+FOXP3+ T cells is associated with HNSCC progression. a** A t-SNE plot shows the T/NK cell clusters including naïve T (IL7R+), CD8+CCL5+, Treg (CD4+FOXP3+), CD4+CD154+, cycling T, and NK cells (*top*). ND, non-determined. Proliferation scores and the number of cells across T cell clusters are shown (*bottom*). **b** Boxplots show the proportion of the T cell clusters according to the tissue types and HPV infection status. Statistical significance of the two-sided *t*-tests is indicated. **c** Distributions of the pseudo-times in each T cell cluster are shown according to the tissue types and HPV infection status. **d** A t-SNE plot shows the *LAIR2* expression in the Tregs (*left*). Stepwise expression of *LAIR2* in Tregs across the tissue types is shown (*right*). **e** Correlation plots show the *COL1A1* expression levels in fibroblasts with the *LAIR2* expression levels in Tregs (*left*) or the proportion of Tregs (*right*) across the tissue types. 95% confidence interval (CI) is indicated with gray color. **f** *FOXP3* expression levels are measured by RT-PCR in the Tregs (see "Methods") treated with or without collagen I (10 µg/mL), respectively.

*n* = 3 biologically independent experiments. Data are shown as mean ± SD. **g** Representative FACS analysis shows the percentages of the FOXP3+CD25+Tregs in PBMCs (*left*). A dot plot shows the percentages of FOXP3+CD25+Tregs in the PBMCs treated with phosphate-buffered saline (PBS) or collagen I, respectively (*right*). *P* = 0.0194. For each group, PBMCs from five patients were evaluated. **h** Representative FACS analysis shows the LAIR2 +FOXP3+Tregs in the PBMCs treated with PBS or collagen, respectively (*left*). A dot plot shows the percentages of LAIR2+FOXP3+CD25+Tregs in the PBMCs treated with PBS or collagen I, respectively (*right*). *P* = 0.0298. In **g**, **h**, for each group, PBMCs from five patients were evaluated. *,*P* < 0.05; **,*P* < 0.01; ***,*P* < 0.001; ns, nonsignificant by the Student two-tailed unpaired *t*-test. In **a**, **b**, **d**, box plots show the median (center line), the upper and lower quantiles (box), and the range of the data (whiskers). Source data are provided as a Source Data file.

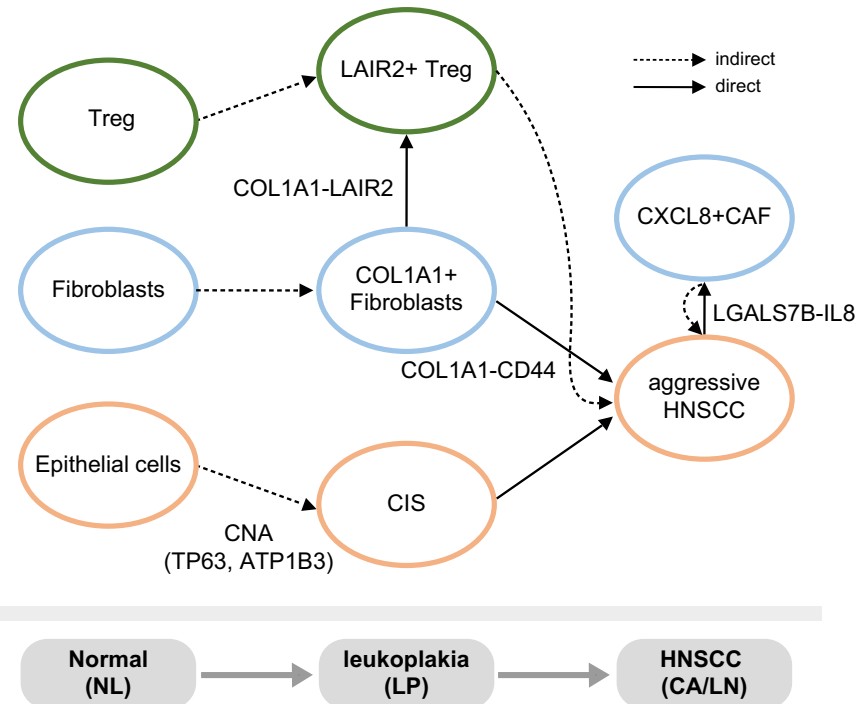

**Fig. 7 | Graphical summary of the cell-cell interactions during stepwise HNSCC progression.** The key findings regarding the interactions among the malignant cells, fibroblasts, and immune cells are summarized.

We demonstrated that the LP tissues had a substantial number of CIS cells which showed DNA copy gains and concomitant transcriptional regulation of *TP63* and *ATP1B3*. Indeed, most CIS cells were identified from the patient P15, showing similar CNAs to those of the malignant cells in CA tissues (Supplementary Fig. 14). It is well-known that severe squamous dysplasia or CIS-causing leukoplakia has a higher rate of progression to invasive carcinoma[35]. Indeed, patient P15 was diagnosed with epithelial dysplasia in 2019, and scRNA-seq was performed at that time. Unfortunately, the patient was re-diagnosed with CIS in 2021. This clinical course of the patient shows that scRNA-seq analysis can help diagnose lesions more accurately and earlier than conventional pathological tests.

In addition, our analyses could identify the cell-cell interactions that promote HNSCC progression. *COL1A1-CD44* signaling mediates the interaction between fibroblasts and malignant cells (see Fig. 4). Moreover, a subset of malignant cells (*LGALS7B*-expressing CC1) interacted with a subset of CAF (*CXCL8*-expressing CF1) (see Fig. 5). From these results, we suggest that *COL1A1-CD44* mediates fibroblast-to-malignant cell interaction, whereas *LGALS7B-CXCL8* mediates malignant cell-to-fibroblast interaction.

Immune cells are essential regulatory components in the tumor microenvironment. We demonstrated stepwise alteration of T cell repertoires during HNSCC progression, implying changes in the immune environment before malignancy. Supporting this, pre-malignant lesions of HNSCC have shown altered inflammatory T cells and increased infiltration of Tregs, promoting tumor progression[36,37]. Notably, we observed that Tregs exhibited transcriptional transition from NL to LP, expressing *LAIR2*. *LAIR2* expression has previously been shown to associate with favorable prognostic outcomes in cancer[38], whereas the contradictory results have also shown that *LAIR2* is a potential marker for T cell exhaustion and worse prognosis in liver cancer[39]. *LAIR2* has a high affinity for various collagen molecules, competing with *LAIR1*-mediated signaling;[34] however, we observed that *LAIR1* expression levels were considerably low in NK/T cells; therefore, they are unlikely to compete with each other (Supplementary Fig. 15). Based on our results, we suggest that the *COL1A1-LAIR2*

interaction between fibroblasts and Tregs provides a permissive microenvironment in tumor development and progression.

In addition to the Tregs, *CD154*-expressing CD4+ T cells were increased during tumor progression (see Fig. 6b). *CD154* is a *CD40* ligand that facilitates cell-cell interactions with the *CD40*-expressing cells, including B cells, macrophages, and follicular helper T cells[40]. *CD154* induces cell proliferation and invasion and poor prognosis, implying that *CD154*+T cells provide a tumor-promoting immune environment[41]. By contrast, cycling T cells build up a dysfunctional program and tumor-infiltrating lymphocyte (TIL) activation[33]. Consistently, we demonstrated that the expression levels of the cycling T cell markers were significantly correlated with the expression levels of dysfunctional T cells markers ($r = 0.47$, $P = 1.5 \times 10^{-28}$) and the TIL scores ($r = 0.3$, $P = 3.66 \times 10^{-12}$, Supplementary Fig. 16). Although we did not further evaluate the functions of CD154+CD4 T cells and cycling T cells, our results imply that the alteration of T cell repertoires plays critical roles in the stepwise progression of HNSCC.

We evaluated the cell proportion in tissues using deconvolution analysis. Cell composition in individual samples can be highly heterogeneous according to the sample preparation and experimental procedures. Therefore, the cell proportion obtained from scRNA-seq may not represent the actual cell proportion in tissues. In this study, we compared the cell composition using the deconvolution analyses of the bulk RNA-Seq data, which could estimate the cellular composition in the tissues. Indeed, we identified a large proportion of malignant cells in deconvolution analyses, although the scRNA-seq data detected only a low proportion of malignant cells. Our analysis shows that in addition to measuring the change in the transcriptional expression of cells, measuring the proportional changes of the cell types can reflect the cancer progression according to the microenvironment alteration.

In conclusion, we provide insights into the single cell-level heterogeneity and interactions during the stepwise progression of HNSCC. Targeting the cell-cell interactions that drive the HNSCC progression would be a crucial strategy for the management of HNSCC.

## Methods

### Patients and tissue specimens

All the experiments with patient samples were performed under the approval of the Ajou University Institutional Review Board, using the approved protocol AJIRB-BMR-SMP-18-150 with the written informed consent of all patients. No compensation was provided for patient participant. A total of 37 tissue specimens were obtained from 23 patients (HNSCC cohorts from Ajou University Hospital), including tissues from NL, LP, CA, and LN.

### scRNA-seq profiling

Single cells in the tissues were obtained using the standard cell dissociation protocols with a Human Tumor Dissociation Kit (Miltenyi Biotec, Bergisch Gladbach, Germany) according to the manufacturer's instructions. Barcoded sequencing libraries were generated using Chromium Single Cell 3′ v2 Reagent Kits and sequenced using the HiSeq 4000 platform (Illumina, CA, USA). Sequencing data were aligned to the human reference genome (GRCh38) and processed using CellRanger 2.1.1 (10X Genomics).

### Cell typing and data analysis

Cell-level transcripts were clustered using the shared nearest neighbor method implemented in the 'Seurat' package. Cell type was determined by examining the expression of the cell type markers, as previously described[21]. The differentially expressed genes for each cluster against the other clusters were determined by a Wilcoxon Rank Sum test with $P$-value <0.001 and fold difference > 1. Immune cells with frequent CNA (>200) were classified as undetermined cells. Further details of the data analyses are described in Supplementary Methods.

### Identification of the recurrent CNA

The DNA copy numbers were inferred using an "inferCNV" and a circular binary segmentation algorithm implemented in the "DNAcopy" R package. Inferring the copy number variations (CNVs) can falsely estimate the CNAs by the chromosomal gene clusters which have similar gene functions. To overcome this limitation, we excluded the gene clusters (within 1 Mb) in which genes were functionally enriched with a gene ontology term ($P < 0.05$). The genes with immune-related RNA biotypes and the genes that reside in sex chromosomes were also excluded. The DNA copy numbers of the immune cells in NL tissues were taken as the reference, then the CNAs were identified with more than a fold difference cutoff of 0.1 compared to the reference. The recurrent CNAs were determined with a cutoff of 0.05 in at least two samples in each tissue type.

### CAF and conditioned medium

The resected tumor tissue samples were immediately cut into small pieces and dissociated into single cells. Fibroblasts were isolated using human anti-fibroblast microbeads (#130-050-601, Miltenyi Biotec) and cultivated in Dulbecco's modified Eagle's Medium/Nutrient Mixture F-12 (DMEM/F12; Welgene Inc., Gyeongsangbuk-do, South Korea). The medium for growing confluent CAFs was changed to 3 mL of serum-free medium in a 100 mm cell culture dish.

### Isolation of CD4+ T cells and generation of Tregs

CD4+ T cells were isolated from C57/BL6 mouse lymph nodes and spleen using mouse CD3+ T cell enrichment columns (R&D Systems, MN, USA) and CD4+MicroBeads (Miltenyi Biotec) according to the manufacturer's instructions. Isolated CD4+ T cells were cultured with anti-CD3 (10 μg/mL), anti-CD28 (10 μg/mL) antibodies (Biogems, USA), mIL-2 (5 μg/mL), and hTGF-β (5 μg/mL) (PeproTech, NJ, USA) for 4 days to induce Tregs. On day 4, the Tregs were collected and cultured on the collagen I-coated plate for 18 h. The cells were harvested and stored at −80 °C until use.

### Flow cytometry of human T cells

Human peripheral blood mononuclear cells (PBMCs, $n = 1 \times 10^7$) were isolated from five HNSCC patients using a SepMate PBMC isolation tube containing Ficoll (STEMCELL Technologies, Canada). The PMBCs were cultured on Collagen type 1 (10 μg/cm²) coated or non-coated plate for 36 h. After cell collection, the cells were stained with monoclonal antibodies for CD3, CD4, CD25, FOXP3 (Invitrogen, Carlsbad, CA), and LAIR2 (R&D system, Minneapolis, MN). Then, the intracellular staining was performed using the Foxp3/Transcription Factor Staining Buffer set (Invitrogen Carlsbad, CA). Flow cytometry was performed using a Becton Dickinson FACSAria III (BD Biosciences, Franklin Lakes, NJ) and analyzed using FlowJo software (Ashland, OR).

### Statistical analysis

Statistical analysis was performed using the R software (version 3.4.0; Vienna, Austria). Survival analyses were performed using Kaplan–Meier plot analyses with log-rank tests.

### Molecular experiments

Cell culture, reagents, immunohistochemistry, western blotting, RT-PCR, siRNA-mediated knockdown, proliferation, migration, invasion, tumor sphere formation, and cell cycle analyses are described in Supplementary Methods. The resources used in the experiments are summarized in Supplementary Table 3.

### Reporting summary

Further information on research design is available in the Nature Portfolio Reporting Summary linked to this article.

## Data availability

The raw and processed data generated in this study have been deposited in the database under accession code GSE181919. The processed single-cell datasets and bulk transcriptome profiles of HNSCC used in the study are obtained from: Puram[21] (scRNA-seq, GSE103322), Kurten[42] (scRNA-seq, GSE164690), Lohavanichbutr[22] (Microarray, GSE41613 and GSE42743), Wichmann 23 (Microarray, GSE65858), and TCGA-HNSCC (RNA-Seq, https://portal.gdc.cancer.gov/). Source data are provided with this paper.

## Code availability

The code that supports the findings of this study is available from the corresponding authors upon request.

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

## Acknowledgements

This research was supported by grants from the National Research Foundation of Korea (NRF) funded by the Korean government (MSIP) (NRF-2018R1A2B3009008, C.H.K. and NRF-2019R1A5A2026045, H.G.W.) and the Korea Health Technology R&D Project through the Korea Health Industry Development Institute (KHIDI) funded by the Korean government (MOHW) (HR21C1003, C.H.K. and H.G.W.). This work was supported by Korea Research Environment Open NETwork (KREONET), managed and operated by Korea Institute of Science and Technology Information (KISTI).

## Author contributions

J.H.C. performed data analyses. B.S.L., Y.S.L., and H.J.K. performed molecular experiments. J.Y.J., J.R., and Y.S.S. acquired tissues and data. H.G.W. and C.H.K. developed the study design and interpreted data and supervised the study. All the authors contributed to writing the manuscript.

## Competing interests

The authors declare no competing interests.
