## [Peer Review File · Nature Communications]

Single-cell transcriptome profiling of the stepwise progression of head and neck cancerREVIEWER COMMENTS

Reviewer #1 (Remarks to the Author): Expert in HNSCC genomics and single-cell RNA-seq

I read “Single-cell transcriptome profiling of the stepwise progression of head and neck cancer” by Choi et al with interest. HNSCC progression is certainly an interesting and important question and I think the authors have a good approach to this problem. Unfortunately, the conclusions are quite vague and rather indeterminate and there are major technical and analytical concerns. The manuscript feels like a mish mash of “one off” observations with no clear story or really validated finding, hampering the novelty and conceptual advance.

1. Association of poor prognosis with a single gene is riddled with errors and issues with false discovery. I am underwhelmed by the idea that LGALS7B and CXCL8 correlate with outcome as it is well established that fibroblasts in general lead to worse outcomes based on the TCGA and other studies.
2. Interaction with CD44+ malignant cells in and of itself if not a mechanism. Furthermore, over the past 2 decades, there has been significant doubt over whether these are truly cancer stem cells given that these properties are not appreciated in patients. For example, do the authors see that such cells are stem cells by their own analysis? I doubt it as no prior HNSCC single cell study has shown that.
3. The numbers are too low to make meaningful conclusions about the relative amount of regulatory T-cells. As the authors likely know, only 10% of leukoplakia lesions progress to cancer, so its unclear what the authors are really trying to support here.
4. LGALS7B has been associated with immune suppression and this is not really seen in this study which would be expected. Also, even if able to show this, showing that there is immune exclusion raises a problem of knowing which is first – did the immune exclusion happen before malignancy or after transformation and what’s the role of this in tumor progression.
5. There is zero genetic (inferred CNV) analysis which is quite surprising as we would expect to see a stepwise progression in this to support the authors methodology.
6. I am hesitant to believe anything about cell proportions because the tumor tissue is likely to be a large piece than the surround normal mucosa or leukoplakia (which are both small by nature) and thus subject to distinct effects with sample processing. Unless using a bulk method, I simply cannot believe any proportions of fibroblasts, epithelial cells, or immune cells. This represents a major concern.
7. I don’t agree that a more epithelial score (2A bottom) shows that cells have acquired a proliferative activity similar to tumors – clinically we know this is not true as these are slow growing lesions, many of which do not transform.
8. It is unclear how many specimens are really paired across all stages and this requires clarification – looks like just a few even though the authors say 23 patients.

9. CNAs – immune cells should not have CNAs, and LN mets should have CNAs. The fact that this is not seen is very worrisome. Also chr 22 is amplified in the leukoplakia but not in the cancer/LN, again this would be something that is never described before – namely, losing a chromosome that’s been amplified and is very worrisome (Fig 2D).

10. Findings related to TP63 and ATP1B3 are not really validated or supported. There is no data to suggest TP63 can transform HNSCC and showing 2 genes is a straw man approach which lacks rigor.

11. Defining 9 malignant clones is very strange – normally one would define these as specific programs based on NMF or other approaches. This is highly unconventional and confusing, and again raises concerns. Same comments for CAFs and immune cells. In general, HPV positive tumors are thought to have less intratumoral heterogeneity (see abundant work from Jim Rocco and others), again raising concerns.

12. LGALS7B was previously identified by another HNSCC single cell analysis (Bernstein/Regev groups) as part of an epithelial differentiation signature. This was shown to correlate with worse prognosis but was part of a large program. The authors have focused on a single gene with no sense of how this affects biology. Identifying single genes from single cell analyses and using those prognosis or to identify cellular heterogeneity is fraught with error. The same can be said of the studies of CXCL8 as again picking one gene in the program and stating that this is the critical driver is very incomplete and troublesome.

13. EMT in fibroblasts?? I don’t believe this has been shown and is a very strange idea. The authors say this “may indicate that EMT in fibroblasts contributes to the malignant conversion of epithelial cells” but fibroblasts are mesenchymal. Just does not make sense, no precedent and no data to support this.

14. The ICC experiments in 4E are strange – I do not see any evidence of colocalization here. There are plenty of CD44 cells that are nowhere near COL1A1 cells and this does not appear to progress with Ca. The tumor is clearly on the left side of these images and I suspect that the CD44 cells in the stroma in the right panel are not tumor cells. Would need to see p63 or cytokeratin to convince me otherwise.

15. Invasion assays have matrigel which is riddled with ECM factors, this makes it a hard system to judge the effects. A better approach would be to co-culture their cell lines and its unclear why this was not done.

16. Correlation of COL1A1 and LAIR2 is weak by eye, does not appear carefully done. Is this true in TCGA deconvolution? That would be the real test.

17. Details of single cell analysis are severely lacking. The computational approaches are essentially absent.

Reviewer #2 (Remarks to the Author): Expert in single-cell RNA-seq, cancer genomics, and immunogenomics

The manuscript presented by Ji-Hye Choi et al. aims to characterize the stepwise progression of HNSCC by applying single-cell RNA-seq technology in normal, leukoplakia, HNSCC and metastasized HNSCC samples. The study contributes with new insights into the molecular mechanisms of stromal, immune and malignant cells that are involved in the non-neoplastic lesion to metastatic tumor transition. Therefore, this work broadens the knowledge of HNSCC stepwise progression and makes it possible to exploit these findings from the therapeutic perspective.

The abstract and introduction point out the relevance for this research, as well as the motivation behind, and they are clearly structured. However, the HPV infection status and its effect in HNSCC progression is not introduced and well explained, neither the motivation behind including this variable in the study (explained in results).

Overall the methods are well explained. However, the manuscript lacks clarity in some parts and could be further improved if authors address the following points.

Could authors make clear if all the tissues representing the stepwise progression of HNSCC come from the same matched patients?

In the paragraph titled “DNA copy number aberrations (CNAs) during HNSCC progression” (row 105), the authors state they could re-assign cells from CA and LN to malignant cells because of their malignant properties. Can authors clarify which are the malignant properties they are referring to? Are they referring to the proliferative activity of those cells?

How did they infer the CNAs? It should be clearly stated in the main text that these have been inferred (and how) directly from the scRNA-seq data.

How did authors obtain clusters in figure 3A? Are these clusters based on their similarity in their transcriptomics profile?

For the identification of CA (cancer-associated)-specific up-regulated gene sets, the fold difference (CA vs. normal tissue) value is different for HPV- and HPV+ cells. Why are these cut-off values chosen and why are they differently selected depending of the HPV infection presence? [Supplementary file, line 31]

In the ligand-receptor analysis, to identify the putative-ligand receptor pairs, the fold difference (ligand vs. receptor) in cancer-associated fibroblast and malignant cells (>0.5) is different to the fibroblast (>0.1). Which is the explanation for this? [Supplementary file, line 55]

For the trajectory analysis, why are 2 different versions of Monocle used in the same analysis for different steps. Why is not the same version used for the whole analysis? [Supplementary file, line 59 and 66]

Reviewer #3 (Remarks to the Author): Expert in tumour microenvironment and stroma, cancer-associated fibroblasts, and immune microenvironment

In the manuscript “Single-cell transcriptome profiling of the stepwise progression of head and neck cancer” the authors employed single-cell RNA-seq of normal, leukoplakia, HNSCC, and metastasized HNSCC. They identified alterations in cell composition with disease stage and clones of malignant cells expressing LGALS7B and fibroblasts expressing CXCL8. These were associated with poor outcome. Authors claim that Col1a produced by fibroblasts interacted with CD44+ tumour cells to support progression, and also supported Treg activation and expansion.

While the authors describe interesting phenomena at the RNA level, the conclusions overstate the data shown. The manuscript would benefit from a more detailed analysis and discussion of the data, showing differentially expressed genes that define key populations as the disease progresses. Validation could be stronger or at least described more thoroughly in the text.

Specific points:

- There are 23 patients in total, but it is not clear how many were from LP vs CA and metastasis i.e. is each of the 23 samples from a different patient or are multiple samples i.e. norm adjacent and cancer and met taken from the same patient? Include a table of patient characteristics or at least disease stage for each sample
- Supp Fig1b contains important information and would be relevant to put in the main figure 1, expanding on the information it shows.
- There seems to be a significant expansion in the endothelial compartment in LP (Fig 1 C). this is not explored in the manuscript. What could underpin this, and does the RNA seq indicate a significant change in phenotype that accompanies a change in abundance?

- What is the proliferation score of immune cells in the diseased tissue? Answering this would provide further biological insight into the phenomenon of TIL activation and response in situ vs recruitment
- Labelling of Fig3A and B could be better or accompanied with more explanation in the legend. I understand there are 8 clone clusters and 3b then shows that LN cluster with the primary tumour, but it is not clear how these are linked to 3A, or how these relate to the patient id; 3A has CC0-8, then 3B has C09, C43 etc. This goes back to the earlier comment of a table detailing patient sample information
- What are the top differentially expressed genes between epi with stage or clone? What factors upregulated in the epithelial compartment in norm vs LP – can then infer interactions with immune cells and fibroblasts
- What are the top differentially expressed genes between HPV+ and – clusters?
- I would not say EMT for fibroblasts – they are already mesenchymal. It may be more appropriate to infer activation status.
- In light of the numerous recent studies that have highlighted significant heterogeneity, the authors should be encouraged to discuss what the differential genes between fibroblast clusters and disease stage are. What are the key gene identifiers of each cluster? These clusters may infer functional specializations
- Supp Fig 6 is difficult to interpret. It is difficult to determine who has the ligand and who has the receptor. Presenting this data as ligand-receptor pairs as shown in for example doi: [10.1016/j.celrep.2020.107628](https://doi.org/10.1016/j.celrep.2020.107628), or doi: [10.1038/s41467-021-22801-0](https://doi.org/10.1038/s41467-021-22801-0)
- Fig.4: CD44 is not co-localized with collagen – this would present as yellow as signals overlap. There are more CD44+ cells within the collagen matrix. As collagen is not a cellular marker, and CD44 can also mark CAFs, additional markers would be needed to confirm if these are tumour or stromal cells to infer anything.
- Migration of CD44 siRNA cells was significantly lower in control conditions. The authors need to quantify any additional effect imparted by CAFs and collagen.

- Fig 5: Are CAF clusters now limited to the tumour samples alone? Does this include LP stages? Or are these solely from published data in Puram. It is not clear to the reader. Again, what are the to differentially expressed genes that distinguish each cluster?

- There is staining for CXCL8 in supp Fig 10a, but without additional markers, the authors cannot conclude what cell population this marks. It looks like immune cells.

- The conclusion at the bottom of page eight is very strong for the data presented. the authors need to perform more detailed interaction analyses between CC1 and CF1 to try and substantiate such a relationship.

- Which T cell populations are proliferating in Figure 6?

- The legend for fig 6 needs to more clearly explain what figure panels show (same for other legends); e.g. detail of what was done in panel and statistics. It is not clear whether validation in 6G is at the RNA or protein level. Protein validation should be included and ideally with functional assays to show Tregs are indeed more activated as the authors suggest.

How many times was the experiment performed? How were iTregs generated, and purity assessed for each replicate? There is no mention of this in the methods, or what concentration of collagen was used.

General comments

- There is a general vagueness in the description and detail of approaches used by the authors.
- There is no indication of statistics used or number of times experiments were performed on validation
- Figure 5G is not referred to in the legend
- Details of antibody concentrations for staining not included
- Details of collagen inclusion or concentrations into invasion assay not included

Point by point replies to the reviewers' comments

Reviewer's Comments:

Reviewer #1:

I read "Single-cell transcriptome profiling of the stepwise progression of head and neck cancer" by Choi et al with interest. HNSCC progression is certainly an interesting and important question and I think the authors have a good approach to this problem. Unfortunately, the conclusions are quite vague and rather indeterminate and there are major technical and analytical concerns. The manuscript feels like a mish mash of "one off" observations with no clear story or really validated finding, hampering the novelty and conceptual advance.

1. Association of poor prognosis with a single gene is riddled with errors and issues with false discovery. I am underwhelmed by the idea that *LGALS7B* and *CXCL8* correlate with outcome as it is well established that fibroblasts in general lead to worse outcomes based on the TCGA and other studies.

Response:

We have demonstrated that the proportions of the malignant cell cluster (CC1) and fibroblast cluster (CF1) are associated with HNSCC prognosis. However, we did not suggest that a single gene is predictive for tumor prognosis. *LGALS7B* and *CXCL8* were the most prominently expressed genes in CC1 and CF1, respectively. These genes have been shown to harbor onco-promoting functions, supporting the aggressive feature of CC1 and CF1 clusters. We validated the aggressive phenotypes of the patients with abundant CC1 or CF1 cells using independent data sets, which supported the robustness of our findings.

2. Interaction with CD44+ malignant cells in and of itself if not a mechanism. Furthermore, over the past 2 decades, there has been significant doubt over whether these are truly cancer stem cells given that these properties are not appreciated in patients. For example, do the authors see that such cells are stem cells by their own analysis? I doubt it as no prior HNSCC single cell study has shown that.

Response:

In the receptor-ligand interaction database, *COL1A1* and *CD44* were the putative ligand-receptor pair. However, their functional interaction between them was not studied thoroughly. In this study, we

demonstrated the interaction between *COL1A1* and *CD44*, using siRNA-mediated knockdown experiments using different systems such as conditioned media and co-culture experiments. These findings strongly suggest the *COL1A1-CD44* interaction.

Previously, many studies have shown that CD44 is a cancer stem cell marker and is associated with aggressive cancer behaviors. However, we did not investigate cancer stem cell-like characteristics of the CD44 expressing cells. In the Discussion section, we explained that these cells might be related to cancer stem cells. However, we removed it from the revised manuscript to clarify the manuscript.

3. The numbers are too low to make meaningful conclusions about the relative amount of regulatory T-cells. As the authors likely know, only 10% of leukoplakia lesions progress to cancer, so its unclear what the authors are really trying to support here.

Response:

We agree with the reviewer's comment that our study has limitations in sample size and the number of cells. Although the sample size was limited, we demonstrated the stepwise pattern of altering T cell repertoires during cancer progression. From the trajectory analysis, we identified that the *LAIR2* expression in Tregs facilitates Treg expansion and tumor progression. Our strategy to identify the stepwise patterns could successfully determine the functional significance of *LAIR2* in Tregs.

We agree that only 10% of the leukoplakia progresses to cancer; thus, the features observed in the leukoplakia lesions may not necessarily be related to the cancer development. However, it is generally agreed that the multiple genetic or environmental aberrations may accumulate during cancer progression, providing tumor-promoting environments. Our and previous studies have suggested that the early events of DNA copy number aberrations with concomitant transcriptional deregulation in the premalignant lesions are likely to play potential driver roles during cancer progression. In addition, we validated the functional significance of our findings by performing cell culture experiments and computational analyses.

4. *LGALS7B* has been associated with immune suppression and this is not really seen in this study which would be expected. Also, even if able to show this, showing that there is immune exclusion raises a problem of knowing which is first – did the immune exclusion happen before malignancy or after transformation and what's the role of this in tumor progression.

Response:

We appreciate the reviewer's comment. Unfortunately, we did not investigate the immunosuppressive roles of *LGALS7B*. However, we assessed whether the CF1 and CC1 abundances are associated because both CF1 and CC1 abundances had aggressive features. As expected, CC1 and CF1 abundances were significantly correlated in the pooled HNSCC data ($r = 0.41$, $P = 1.11 \times 10^{-38}$, **Fig. 5f**). In addition, we demonstrated that galectin-7 treatment significantly induced IL-8 expression in both HNSCC and CAF cells (*i.e.*, CAF30, CAF57, CAF58, and CAF70, **Fig 5g**). These results suggest that galectin-7 activates both malignant cells (CC1) and fibroblasts (CF1) to express IL-8, resulting in aggressive progression of HNSCC. We added this data in the revised manuscript.

5. There is zero genetic (inferred CNV) analysis which is quite surprising as we would expect to see a stepwise progression in this to support the authors methodology.

Response:

Previously, genetic DNA copy number alterations in 3p, 3q, and 8p have already been reported, which we could confirm in our data. Following the reviewer's comment, we validated our findings using the HPV-negative TCGA-HNSCC data, showing significant correlations between DNA copy numbers and transcription levels of *TP63* ($r = 0.19$, $P = 2.5 \times 10^{-3}$) and *ATP1B3* ($r = 0.37$, $P = 2.0 \times 10^{-9}$, **Fig. 2f**). We also validated the functions of *TP63* and *ATP1B3* by performing siRNA-mediated knockdown experiments. Treatment with siRNAs targeting *TP63* (si*TP63*) or *ATP1B3* (si*ATP1B3*) suppressed tumor-promoting functions of HNSCC cells, including proliferation, migration, invasion, and sphere formation (**Fig. 2g, h, i**). Thus, we suggest that the CNA-dependent expression of *TP63* and *ATP1B3* plays a critical role in HNSCC progression. We added the data in the revised manuscript.

6. I am hesitant to believe anything about cell proportions because the tumor tissue is likely to be a large piece than the surround normal mucosa or leukoplakia (which are both small by nature) and thus subject to distinct effects with sample processing. Unless using a bulk method, I simply cannot believe any proportions of fibroblasts, epithelial cells, or immune cells. This represents a major concern.

Response:

We agree with the reviewer's comments. Cell composition in individual samples can be highly heterogeneous according to the sample preparation and experimental procedures. We prepared the

large piece from the tumor tissues, but only a small proportion of the cells were profiled. We agree that cell proportion obtained from scRNA-seq may not represent the actual proportion in tissues. With this concern, we compared the cell composition using deconvolution analyses of the bulk RNA-Seq data. Indeed, only a small number of malignant cells were detected in the scRNA-seq data. However, deconvolution analysis of bulk RNA-seq showed that malignant cells were the most common (79.17%), followed by fibroblasts (10.07%) (**Supplementary Fig. 4a**). We added this statement in the revised manuscript.

7. I don't agree that a more epithelial score (2A bottom) shows that cells have acquired a proliferative activity similar to tumors – clinically we know this is not true as these are slow growing lesions, many of which do not transform.

Response:

We greatly appreciate the reviewer's comment. We found that the data was confounded with HPV infection status; thus, we re-analyzed the epithelial and proliferation scores of the epithelial/malignant cells according to the HPV infection status. We found that the HPV-negative LP tissues had higher epithelial scores than those of NL tissues, implying the transformation of the epithelial cells in LP tissue ($P < 5.03 \times 10^{-4}$, NL vs. LP, **Fig. 2a, left**). However, the proliferation scores of the LP tissues were not much altered compared to those of the NL tissues, which agrees with the slow-growing characteristics of the LP tissues (**Fig. 2a, right**). We added these data and statements in the revised manuscript.

8. It is unclear how many specimens are really paired across all stages and this requires clarification – looks like just a few even though the authors say 23 patients.

Response:

We apologize for any inconvenience caused by the insufficient sample information. We added the details on the pair-matched sample information and the patient's clinicopathological information in **Supplementary Table 1**.

9. CNAs – immune cells should not have CNAs, and LN mets should have CNAs. The fact that this is not seen is very worrisome. Also chr 22 is amplified in the leukoplakia but not in the cancer/LN, again this

would be something that is never described before – namely, losing a chromosome that's been amplified and is very worrisome (Fig 2D).

Response:

We appreciate the reviewer's valuable comment. The T cells having CNAs (449 of 17,869 cells, 2.51%) were re-assigned as the undetermined cells and removed in the analysis.

We observed the DNA copy gain at chr22 in CA and LN tissues (C30, C38, C15, and LN38); however, these CNAs were only observed in a few cells (3.92 %). This might be due to the heterogeneous cell proportion across the samples. To overcome this limitation, we modified the CNA calling method to detect the recurrent CNAs, considering the different cell proportions in each sample. We determined the tissue type-specific CNAs and the sample-specific CNAs, respectively. Tissue-specific CNAs were determined by comparing the frequencies of the cells with CNAs in each tissue (*i.e.*, LP vs. NL and CA/LN vs. LP) with a cutoff of the fold difference of the cell proportion with CNAs greater than 10 % and Fisher's exact test ($P < 0.001$) between the tissues. In addition, the sample-specific CNAs in the epithelial cells were determined with a cutoff of a cell proportion with CNA greater than 5 %. Then, we defined the recurrent CNAs as the genes observed in at least two samples with the overlap between the tissue-specific and the sample-specific CNAs. We described this statement in the revised **Supplementary Methods**.

10. Findings related to TP63 and ATP1B3 are not really validated or supported. There is no data to suggest TP63 can transform HNSCC and showing 2 genes is a straw man approach which lacks rigor.

Response:

Following the reviewer's comment, we validated our findings using the HPV-negative TCGA-HNSCC data, showing significant correlations between DNA copy numbers and transcription levels of *TP63* ($r = 0.19$, $P = 2.5 \times 10^{-3}$) and *ATP1B3* ($r = 0.37$, $P = 2.0 \times 10^{-9}$, **Fig. 2f**). We also validated the functions of *TP63* and *ATP1B3* by performing siRNA-mediated knockdown experiments. Treatment with siRNAs targeting *TP63* (si*TP63*) or *ATP1B3* (si*ATP1B3*) suppressed tumor-promoting functions of HNSCC cells, including proliferation, migration, invasion, and sphere formation (**Fig. 2g, h, i**). Thus, we suggest that the CNA-dependent expression of *TP63* and *ATP1B3* plays a critical role in HNSCC progression. We added the data in the revised manuscript.

11. Defining 9 malignant clones is very strange – normally one would define these as specific programs based on NMF or other approaches. This is highly unconventional and confusing, and again raises concerns. Same comments for CAFs and immune cells. In general, HPV positive tumors are thought to have less intratumoral heterogeneity (see abundant work from Jim Rocco and others), again raising concerns.

Response:

We appreciate the reviewer's valuable comment. We have identified the malignant cell clusters representing the transcriptional clusters, not the genetic clones. Although we have intended to describe the 'intratumoral transcriptional heterogeneity, it is confusing with the previous term 'intratumoral clonal heterogeneity'. Data of genetic variants might be required to evaluate the association of HPV infection with intra-tumoral clonal heterogeneity. Thus, we corrected the term 'malignant clone' to 'malignant clusters' in the revised manuscript. To make the manuscript more concise, we removed the 'intra-tumoral heterogeneity (ITH)' data in the revised manuscript.

We re-analyzed the malignant cell clusters to identify the clusters showing distinct expression of the marker genes, revealing six malignant cell clusters (CC0 to CC5, **Fig. 3a, top**, see **Supplementary Methods**). We used the cluster algorithms implemented in 'Seurat' package, which is now generally used for the clustering analysis of single cell data.

12. *LGALS7B* was previously identified by another HNSCC single cell analysis (Bernstein/Regev groups) as part of an epithelial differentiation signature. This was shown to correlate with worse prognosis but was part of a large program. The authors have focused on a single gene with no sense of how this affects biology. Identifying single genes from single cell analyses and using those prognosis or to identify cellular heterogeneity is fraught with error. The same can be said of the studies of *CXCL8* as again picking one gene in the program and stating that this is the critical driver is very incomplete and troublesome.

Response:

We agree with the reviewer's comment. We identified *LGALS7B* and *CXCL8* as expression markers for the abundance of CC1 and CF1, respectively. We demonstrated that the proportions of CC1 and CF1 in tumor tissues were associated with an unfavorable prognosis of HNSCCs. Although *LGALS7B* and *CXCL8* were not identified as prognostic markers, tumor-promoting functions of these genes have been shown previously, supporting our findings. Thus, we suggest that the increased proportion of

CC1 and CF1 clusters may contribute to the aggressive progression of HNSCC.

13. EMT in fibroblasts?? I don't believe this has been shown and is a very strange idea. The authors say this "may indicate that EMT in fibroblasts contributes to the malignant conversion of epithelial cells" but fibroblasts are mesenchymal. Just does not make sense, no precedent and no data to support this.

Response:

We greatly appreciate the reviewer's valuable comments. We mis-interpreted the Hallmark gene set analysis result. As we have shown, we observed that EMT gene sets were stepwisely expressed in fibroblasts but not in malignant cells during cancer progression. Previously, the cells undergone EMT were very similar to cancer-associated fibroblasts (CAF), expressing many mesenchymal or EMT markers (Pastushenko et al. *Trends Cell Biol.* 29, 212–226, 2019). A recent study has also shown that the well-defined EMT signatures coexist with the EMT programs of cancer cells and CAF programs, suggesting the need to decouple the mesenchymal expression profiles of cancer cells and CAFs (Tyler et al. *Nature Comm*, 2592, 2021). Indeed, the EMT-related genes included many CAF-related genes (e.g., *POSTN*, *GREM1*, *STHRC1*, *ADAM12*, and *COL11A1*, **Supplementary Fig. 7, right**); thus, we suggest that EMT expression in fibroblasts may indicate the acquisition of CAF-like traits. We replaced this data and interpretation in the revised manuscript.

14. The ICC experiments in 4E are strange – I do not see any evidence of colocalization here. There are plenty of CD44 cells that are nowhere near COL1A1 cells and this does not appear to progress with Ca. The tumor is clearly on the left side of these images and I suspect that the CD44 cells in the stroma in the right panel are not tumor cells. Would need to see p63 or cytokeratin to convince me otherwise.

Response:

We appreciate for reviewer's comments. We replaced the images with a new one, which showed the expression of CD44 in malignant cells and the enhanced expression of COL1A1 in the nearby located fibroblasts. Following the reviewer's comment, we have demonstrated the localized expression of CD44 and the p63 in the malignant cells (**Supplementary Fig. 8a**).

15. Invasion assays have matrigel which is riddled with ECM factors, this makes it a hard system to

judge the effects. A better approach would be to co-culture their cell lines and its unclear why this was not done.

Response:

Following the reviewer's comment, we evaluated the effects of the siRNAs in the co-culture system of the malignant cells and CAFs. We observed that co-culture of MSKQLL1 and CAF48 increased the tumor cell migration. However, the co-culture of si*CD44*-transfected MSKQLL1 cells or si*COL1A1*-transfected CAF48 cells did not enhance the tumor cell migration (**Fig. 4h**). These results suggest that the *COL1A1-CD44* interaction between fibroblasts and malignant cells plays a crucial role in HNSCC progression. We added these data and statements in the revised manuscript.

16. Correlation of COL1A1 and LAIR2 is weak by eye, does not appear carefully done. Is this true in TCGA deconvolution? That would be the real test.

Response:

We demonstrated that *COL1A1* expression levels in fibroblasts were significantly correlated with *LAIR2* expression levels ($r = 0.48$, $P = 2.48 \times 10^{-3}$) and the abundance of Tregs ($r = 0.38$, $P = 2.60 \times 10^{-2}$, **Fig. 6e**). In addition, collagen treatment significantly induced *FOXP3* expression in Tregs (**Fig. 6f**). FACS analysis also confirmed that collagen treatment increased the number of Tregs and LAIR2+ Tregs (**Fig. 6g, h**). Thus, we suggest that the collagen released from the fibroblasts induces Tregs to express *LAIR2*, providing a favorable microenvironment for tumor progression. We added these data and statements in the revised manuscript.

Unfortunately, we could not perform deconvolution analysis with the T cells in TCGA data, because the proportion of the deconvolved T cells was very low.

17. Details of single cell analysis are severely lacking. The computational approaches are essentially absent.

Response:

We apologize for lacking detailed information on data analysis. We entirely and carefully re-wrote the details on the computational analysis methods in **Supplementary Methods**. To guarantee the reproducibility of our results, we also submitted the source data used in the manuscript.

Reviewer #2 (Remarks to the Author): Expert in single-cell RNA-seq, cancer genomics, and immunogenomics

The manuscript presented by Ji-Hye Choi et al. aims to characterize the stepwise progression of HNSCC by applying single-cell RNA-seq technology in normal, leukoplakia, HNSCC and metastasized HNSCC samples. The study contributes with new insights into the molecular mechanisms of stromal, immune and malignant cells that are involved in the non-neoplastic lesion to metastatic tumor transition. Therefore, this work broadens the knowledge of HNSCC stepwise progression and makes it possible to exploit these findings from the therapeutic perspective.

1. The abstract and introduction point out the relevance for this research, as well as the motivation behind, and they are clearly structured. However, the HPV infection status and its effect in HNSCC progression is not introduced and well explained, neither the motivation behind including this variable in the study (explained in results).

Response:

We appreciate the reviewer's comments. Indeed, various etiological factors are involved in HNSCC development, including exposure to alcohol or tobacco and human papillomavirus (HPV) infection. In particular, HPV infection has played a critical role in HNSCC progression. HPV-positive patients showed more favorable prognostic outcomes than the HPV-negative patients. Genomic analyses have shown that the innate and acquired antiviral immune responses are suppressed in HPV-positive patients. Considering the mechanistic and clinical impacts of HPV, we carefully analyzed the single cell transcripts according to the HPV infection status. We added this statement in the Introduction section.

Overall the methods are well explained. However, the manuscript lacks clarity in some parts and could be further improved if authors address the following points.

2. Could authors make clear if all the tissues representing the stepwise progression of HNSCC come from the same matched patients?

Response:

We apologize for any inconvenience caused by the insufficient sample information. We added the details on the pair-matched sample information and the patient's clinicopathological information in **Supplementary Table 1**.

3. In the paragraph titled “DNA copy number aberrations (CNAs) during HNSCC progression” (row 105), the authors state they could re-assign cells from CA and LN to malignant cells because of their malignant properties. Can authors clarify which are the malignant properties they are referring to? Are they referring to the proliferative activity of those cells?

Response:

We first assigned the epithelial cells from the whole cell analysis; then, the epithelial cells in CA and LN tissues were re-assigned as the malignant cells. Indeed, the CNAs observed in the epithelial cells may indicate the presence of carcinoma in situ (CIS) cells in LP tissues, which we assigned as CIS cells. We evaluated the CNA scores and proliferation scores, which showed a significant upregulation in CA/LN tissues than NL or LP tissues, indicating their malignant property. To clarify the description, we added this statement in the revised manuscript.

4. How did they infer the CNAs? It should be clearly stated in the main text that these have been inferred (and how) directly from the scRNA-seq data.

Response:

DNA copy numbers were inferred using the “inferCNV” R package, with modifications to remove the chromosomal transcription by gene clusters. We described the details of the modified inferCNV method in the Supplementary Methods because the description was lengthy.

The gene clusters located in a chromosomal region with similar gene functions can be falsely estimated to have DNA copy number aberrations by the inferCNV method. With this concern, we determined the functional gene clusters by calculating the functional associations of the genes located within a chromosomal region of 1 Mb using gene ontology analysis ($P < 0.05$). Before inferring the DNA copy numbers, we removed the genes residing in the putative functional gene clusters. The immune-related RNA biotypes and sex chromosomes were also excluded from the analysis. Then, the gene-level DNA copy numbers in each cell were estimated by the segmented CNA values using the circular binary segmentation algorithm implemented in the “DNACopy” R package. We added this statement in the Supplementary Methods.

5. How did authors obtain clusters in figure 3A? Are these clusters based on their similarity in their

transcriptomics profile?

Response:

We sub-classified the epithelial cells of CA and LN tissues, revealing six malignant cell clusters (CC0 to CC5, **Fig. 3a, top**, for details, see **Supplementary Methods**). The clusters were obtained based on the transcriptional similarity of the cells using a shared nearest neighbor (SNN) modularity optimization-based clustering algorithm. We added this statement in the revised manuscript.

6. For the identification of CA (cancer-associated)-specific upregulated gene sets, the fold difference (CA vs. normal tissue) value is different for HPV- and HPV+ cells. Why are these cutoff values chosen and why are they differently selected depending of the HPV infection presence? [Supplementary file, line 31]

Response:

We appreciate the reviewer's valuable comments. We re-analyzed the data applying the same cutoff values (>0.1) for the HPV-positive and HPV-negative tumors. We replaced the data in the revised manuscript.

7. In the ligand-receptor analysis, to identify the putative-ligand receptor pairs, the fold difference (ligand vs. receptor) in cancer-associated fibroblast and malignant cells (>0.5) is different to the fibroblast (>0.1). Which is the explanation for this? [Supplementary file, line 55]

Response:

We identified putative interactions between malignant cells and fibroblasts based on the expression levels of the ligand-receptor pairs. Data for putative ligand-receptor interactions were obtained from a previous study⁴. First, the average expression levels for each ligand-receptor pair were estimated in the malignant cells and the fibroblasts across the tissue types of NL, LP, and CA/LN, respectively. Then, to determine the interdependent ligand-receptor pairs between CAFs and malignant cells, we filtered the ligand-receptor pairs using the following criteria. (1) The ligands were expressed, but their receptors were not expressed in CAF (ligand vs. receptor, fold difference of average expression > 0.5). (2) The receptors were expressed, but their ligands were not expressed in the malignant cells (receptor vs. ligand, fold difference of average expression > 0.5). (3) The ligands were expressed in LP than NL (LP vs. NL, fold difference of average expression > 0.1). Because the basal expression levels of the

ligands and the receptors were highly different according to the tissue types, showing higher expression in tumors tissues (CA and LN) than non-tumor tissues (NL and LP), therefore, we applied different cutoff values to determine the fold differences.

8. For the trajectory analysis, why are 2 different versions of Monocle used in the same analysis for different steps. Why is not the same version used for the whole analysis? [Supplementary file, line 59 and 66]

Response:

We performed the pseudo-time trajectory analyses for T cells using Monocle v2 (**Fig. 6c**). In **Fig. 4**, we performed a trajectory analysis with the UMAP-based dimension reduction and single-cell state transition analysis by Monocle version 3, because UMAP-based plot was not supported by Monocle version 2 (**Fig. 4a, right**). We could obtain a similar result of pseudo-time trajectory analysis using Monocle version 2 as shown in the following plot. Thus, we suggest that our findings were not specific to the Monocle software versions.

Reviewer #3 (Remarks to the Author): Expert in tumour microenvironment and stroma, cancer-associated fibroblasts, and immune microenvironment

In the manuscript “Single-cell transcriptome profiling of the stepwise progression of head and neck cancer” the authors employed single-cell RNA-seq of normal, leukoplakia, HNSCC, and metastasized HNSCC. They identified alterations in cell composition with disease stage and clones of malignant cells expressing LGALS7B and fibroblasts expressing CXCL8. These were associated with poor outcome. Authors claim that Col1a produced by fibroblasts interacted with CD44+ tumour cells to support progression, and also supported Treg activation and expansion. While the authors describe interesting phenomena at the RNA level, the conclusions overstate the data shown. The manuscript would benefit from a more detailed analysis and discussion of the data, showing differentially expressed genes that define key populations as the disease progresses. Validation could be stronger or at least described more thoroughly in the text.

Specific points:

1. There are 23 patients in total, but it is not clear how many were from LP vs CA and metastasis i.e. is each of the 23 samples from a different patient or are multiple samples i.e. norm adjacent and cancer and met taken from the same patient? Include a table of patient characteristics or at least disease stage for each sample

Response:

We apologize for any inconvenience caused by the insufficient sample information. We added the details on the pair-matched sample information and the patient's clinicopathological information in **Supplementary Table 1**.

2. Supp Fig1b contains important information and would be relevant to put in the main figure 1, expanding on the information it shows.

Response:

We appreciate the reviewer's suggestion. We replaced **Supplementary Fig. 1b** to **Fig. 1c**.

3. There seems to be a significant expansion in the endothelial compartment in LP (Fig 1 C). this is not explored in the manuscript. What could underpin this, and does the RNA seq indicate a significant change in phenotype that accompanies a change in abundance?

Response:

We greatly appreciate the reviewer's comment. The increased proportion of the endothelial cells in LP might be due to the increased angiogenesis during NL to LP transition, as described previously (Thiem et al. *J Oral Pathol Med.*, 46(9), 710–716, 2017). However, when we evaluated the pair-matched samples (N15 vs. LP15), no significant increase in the endothelial cells was observed. Thus, to avoid any possible mis-interpretation of the data, we removed the data **Fig. 1c**, which showed only the average cell composition from the multiple samples in each tissue type. Instead, we replaced them with the data showing the cell composition for each sample (**Supplementary Fig. 1b**).

4. What is the proliferation score of immune cells in the diseased tissue? Answering this would provide further biological insight into the phenomenon of TIL activation and response in situ vs recruitment

Response:

Previously, cycling T cells have been addressed to be highly proliferative, and build up the dysfunctional program and tumor-infiltrating lymphocyte (TIL) activation (Li et al. *cell*, 176, 775-789, 2019). Consistently, we could demonstrate that the expression levels of the cycling T cell markers were significantly correlated with the expression of the dysfunctional T cells markers and the TIL scores, respectively (**Supplementary Fig. 15**). Although we did not further elaborate on the functions of cycling T cells, our results suggest that the alteration of the T cell repertoires, including cycling T cells, may contribute to the stepwise progression of HNSCC. We added the data in the revised manuscript.

5. Labelling of Fig3A and B could be better or accompanied with more explanation in the legend. I understand there are 8 clone clusters and 3b then shows that LN cluster with the primary tumour, but it is not clear how these are linked to 3A, or how these relate to the patient id; 3A has CC0-8, then 3B has C09, C43 etc. This goes back to the earlier comment of a table detailing patient sample information

Response:

We appreciate the reviewer's comments. **Fig. 3a** was marked by the labels for malignant cell cluster (CC0-CC5), while **Fig. 3b** was marked by the sample IDs (e.g., C09), respectively. We described the details in the Figure legends.

6. What are the top differentially expressed genes between epi with stage or clone? What factors upregulated in the epithelial compartment in norm vs LP – can then infer interactions with immune cells and fibroblasts

Response:

We appreciate the reviewer's valuable comments. We added the data showing the differentially expressed genes (DEGs) for the malignant cell clusters in **Supplementary Fig. 5a**.

We also identified the stepwise expression genes in each cell type during progression from NL to LP and CA tissue. We identified 17 genes (e.g., *KRT6A*) in epithelial/malignant cells and 41 genes (e.g., *POSTN*) in fibroblasts (**Supplementary Fig. 6**). We added these data in the revised manuscript.

7. What are the top differentially expressed genes between HPV+ and – clusters?

Response:

We have stratified the malignant cells into six clusters, which were closely associated with HPV infection status. We identified differentially expressed genes in the malignant cell clusters, revealing the most prominent expression of *CAML1* and *LGALS7B* (galectin-7B) in CC0 and CC1, respectively (Wilcoxon Rank Sum test $P < 0.001$, fold difference > 1 , **Fig. 3g** and **Supplementary Fig. 5a**).

The results suggest that *CAML1* and *LGALS7B* are the top expressed genes for HPV-positive and HPV-negative tumor clusters, respectively. However, we did not further investigate the DEGs and their roles between HPV-positive and HPV-native tumors.

8. I would not say EMT for fibroblasts – they are already mesenchymal. It may be more appropriate to infer activation status.

Response:

We greatly appreciate the reviewer's valuable comments. We mis-interpreted the Hallmark gene set analysis result. As we have shown, we observed that EMT gene sets were stepwisely expressed in fibroblasts but not in malignant cells during cancer progression. Previously, the cells undergone EMT were very similar to cancer-associated fibroblasts (CAF), expressing many mesenchymal or EMT markers (Pastushenko et al. *Trends Cell Biol.*, 29, 212–226, 2019). A recent study has also shown that the well-defined EMT signatures coexist with the EMT programs of cancer cells and CAF programs, suggesting the need to decouple the mesenchymal expression profiles of cancer cells and CAFs (Tyler et al. *Nature Comm.*, 2592, 2021). Indeed, we observed that the EMT-related genes included many CAF-related genes (e.g., *POSTN*, *GREM1*, *STHRC1*, *ADAM12*, and *COL11A1*), indicating the expression of the CAF-like trait in fibroblasts during HNSCC progression (**Supplementary Fig. 7, right**). We replaced this data and interpretation in the revised manuscript.

9. In light of the numerous recent studies that have highlighted significant heterogeneity, the authors should be encouraged to discuss what the differential genes between fibroblast clusters and disease stage are. What are the key gene identifiers of each cluster? These clusters may infer functional specializations.

Response:

We appreciate the reviewer's suggestion. We analyzed the DEGs for the fibroblast clusters. The

fibroblasts expressed many CAF programs during malignant conversion, which was shown in **Supplementary Fig. 7**. In addition, we analyzed the heterogeneity of the CAFs. The CAFs were further stratified into five clusters (CF0 to CF4), revealing their representative marker genes (**Fig. 5a** and **Supplementary Fig. 9a**). When we compared the CAF clusters with the previous CAFs identified from Puram's study ¹², we found that CF0 and CF4 were identical to the Puram's subtype CAF1 expressing its marker genes (*e.g.*, *CTHRC1*, *COL1A1*, *COL3A1*, *POSTN*, and *MFAP2*) (**Fig. 5b** and **Supplementary Fig. 9a**); however, CF4 was distinguished from CF0, showing its higher proliferation activity (**Supplementary Fig. 9b**). CF2 and CF3 were identical to the Puram's CAF2 and myofibroblast, expressing its marker genes such as *CXCL12* and *NDUFA4L2*, respectively. Notably, we identified a novel CAF subtype, CF1, expressing *CXCL8* (IL-8 coding gene). CF1 showed distinct expression patterns among the CAF clusters, showing a unique branch in the two-dimensional diffusion map embedding analysis ³⁰ (**Fig. 5c, left**).

We added these statements in the revised manuscript.

10. Supp Fig 6 is difficult to interpret. It is difficult to determine who has the ligand and who has the receptor. Presenting this data as ligand-receptor pairs as shown in for example doi: 10.1016/j.celrep.2020.107628., or doi: 10.1038/s41467-021-22801-0

Response:

Following the reviewer's suggestion, we added the ligand-receptor information in **Fig. 4c**. Because we sought to identify the interdependent expression of the ligand-receptor pairs between fibroblasts and malignant cells, we want to demonstrate the expression levels of the ligand and receptors in both fibroblasts and malignant cells together. The details on the analysis method used for identifying the interdependent ligand-receptor expression were described in **Supplementary Methods**.

11. Fig.4: CD44 is not co-localized with collagen – this would present as yellow as signals overlap. There are more CD44+ cells within the collagen matrix. As collagen is not a cellular marker, and CD44 can also mark CAFs, additional markers would be needed to confirm if these are tumour or stromal cells to infer anything.

Response:

We appreciate for reviewer's comments. We replaced the images with a new one, which showed the

expression of CD44 in malignant cells and the enhanced expression of COL1A1 in the nearby located fibroblasts. Following the reviewer's comment, we have demonstrated the localized expression of CD44 and TP63 in the malignant cells (**Supplementary Fig. 8a**).

12. Migration of CD44 siRNA cells was significantly lower in control conditions. The authors need to quantify any additional effect imparted by CAFs and collagen.

Response:

We appreciate for reviewer's comments. We added the quantitated results in the revised manuscripts (**Fig. 4h**). The co-culture of MSKQLL1 and CAF48 increased the tumor cell migration; however, the co-culture of siCD44-transfected MSKQLL1 cells or siCOL1A1-transfected CAF48 cells did not. We added these data in the revised manuscript.

13. Fig 5: Are CAF clusters now limited to the tumour samples alone? Does this include LP stages? Or are these solely from published data in Puram. It is not clear to the reader. Again, what are the differentially expressed genes that distinguish each cluster?

Response:

Fig. 5 shows the CAF clusters identified CA and LN tissues not the NL or LP tissues. To clarify the description, we revised the statement as follows. "The CAFs from tumor tissues (CA and LN) were further stratified into five clusters (CF0 to CF4), revealing their representative marker genes (**Fig. 5a** and **Supplementary Fig. 9a**). When we compared the CAF clusters with the previous CAFs identified from Puram's study, we found that CF0 and CF4 were identical to the Puram's subtype CAF1 expressing its marker genes (*e.g.*, *CTHRC1*, *COL1A1*, *COL3A1*, *POSTN*, and *MFAP2*) (**Fig. 5b** and **Supplementary Fig. 9a**); however, CF4 was distinguished from CF0, showing its higher proliferation activity (**Supplementary Fig. 9b**). CF2 and CF3 were identical to the Puram's CAF2 and myofibroblast, expressing its marker genes such as *CXCL12* and *NDUFA4L2*, respectively. Notably, we identified a novel CAF subtype, CF1, expressing *CXCL8* (IL-8 coding gene).

We added these statements and data in the revised manuscript.

14. There is staining for CXCL8 in supp Fig 10a, but without additional markers, the authors cannot

conclude what cell population this marks. It looks like immune cells.

Response:

We appreciate the reviewer's comments. Unfortunately, although we have tried to re-examine the expression of IL-8 in fibroblasts, we could not obtain satisfying results. IL-8 is a secretory molecule and only the smaller proportion of the CAFs express the IL-8; therefore, it might not be easy to observe a quantifiable data. Thus, we removed the data in the revised manuscript.

In addition, we demonstrated that the CAFs induce IL-8 expression by treatment of galectin-7 (**Fig. 5g**). From this result, we suggested the interaction of the LGALS7B-expressing malignant cell clusters may activate CAFs. We added this data in the revised manuscript.

15. The conclusion at the bottom of page eight is very strong for the data presented. the authors need to perform more detailed interaction analyses between CC1 and CF1 to try and substantiate such a relationship.

Response:

We appreciate the reviewer's valuable comments. we assessed whether the CF1 and CC1 abundances are associated because both CF1 and CC1 abundances had aggressive features. As expected, CC1 and CF1 abundances were significantly correlated in the pooled HNSCC data ($r = 0.41$, $P = 1.11 \times 10^{-38}$, **Fig. 5f**). In addition, we demonstrated that galectin-7 treatment significantly induced IL-8 expression in both HNSCC and CAF cells (*i.e.*, CAF30, CAF57, CAF58, and CAF70, **Fig 5g**). These results suggest that galectin-7 activates both malignant cells (CC1) and fibroblasts (CF1) to express IL-8, resulting in the aggressive progression of HNSCC.

Following the reviewer's comments, we have performed ligand-receptor interaction analysis between CF1 and CC1. However, we could not validate the identified interactions experimentally. This might be because the analysis was not performed with a sufficient number of cells to obtain reliable results. Therefore, we did not include this data in the manuscript.

16. Which T cell populations are proliferating in Figure 6?

Response:

We revised the **Fig. 6a** as follows. Cells with chromosomal aberrations were re-assigned as the non-

determinant cell type and excluded (for details, see **Methods**). Then, the NK/T cell cluster was further stratified into six subclusters; naïve T cells, CD8+CCL5+, CD4+FOXP3+ (regulatory T cells, [Treg]), CD4+CD154+ (follicular helper T cells), cycling T cells, and NK cells (**Fig. 6a, top**, and **Supplementary Fig. 11**). The T cell cluster showing higher proliferation activity was designated as a cycling T cell as previously described³¹ (**Fig. 6a, bottom**). We added this statement in the revised manuscript.

17. The legend for fig 6 needs to more clearly explain what figure panels show (same for other legends); e.g. detail of what was done in panel and statistics. It is not clear whether validation in 6G is at the RNA or protein level. Protein validation should be included and ideally with functional assays to show Tregs are indeed more activated as the authors suggest. How many times was the experiment performed? How were iTregs generated, and purity assessed for each replicate? There is no mention of this in the methods, or what concentration of collagen was used.

Response:

We appreciate the reviewer's valuable comments. Following the reviewer's comment, we added the FACS data, demonstrating LAIR2 protein expression in Tregs. We demonstrated that *COL1A1* expression levels in fibroblasts were significantly correlated with *LAIR2* expression levels ($r = 0.48$, $P = 2.48 \times 10^{-3}$) and the abundance of Tregs ($r = 0.38$, $P = 2.60 \times 10^{-2}$, **Fig. 6e**). In addition, we demonstrated that collagen treatment significantly induced *FOXP3* expression in Tregs (**Fig. 6f**). FACS analysis also confirmed that collagen treatment increased the number of LAIR2-expressing Tregs (**Fig. 6g, h**). We added these data in the revised manuscript.

General comments

- There is a general vagueness in the description and detail of approaches used by the authors.

Response: We appreciate the reviewer's comment. We carefully revised the manuscript thoroughly, adding the details of the analysis methods in the supplementary Methods and Figure legends.

- There is no indication of statistics used or number of times experiments were performed on validation

Response: We added the statistics and the number of replicated experiments in the revised figure

legends.

- Figure 5G is not referred to in the legend

Response: We corrected the Figure numbers in the revised manuscript.

- Details of antibody concentrations for staining not included

Response: We added the antibody concentrations in **Supplementary Methods** in the revised manuscript.

- Details of collagen inclusion or concentrations into invasion assay not included

Response: We added the collagen concentration (1 $\mu\text{g}/\text{ml}$) in the revised manuscript.

REVIEWER COMMENTS

Reviewer #1 (Remarks to the Author):

I re-reviewed “Single-cell transcriptome profiling of the stepwise progression of head and neck cancer” by Choi et al. Unfortunately, the conclusions remain very vague, lack novelty, and technical and analytical concerns persist. The authors do not address this major concern.

1. Association of poor prognosis with a single gene is riddled with errors and issues with false discovery. I am underwhelmed by the idea that LGALS7B and CXCL8 correlate with outcome as it is well established that fibroblasts in general lead to worse outcomes based on the TCGA and other studies.

> The authors miss the point here – they are using a few genes to define a cluster as aggressive without a clear biological pathway or underpinning to define this cluster. These analyses therefore are not robustly defined, and the validation in another very small dataset is not convincing.

2. Interaction with CD44+ malignant cells in and of itself if not a mechanism. Furthermore, over the past 2 decades, there has been significant doubt over whether these are truly cancer stem cells given that these properties are not appreciated in patients. For example, do the authors see that such cells are stem cells by their own analysis? I doubt it as no prior HNSCC single cell study has shown that.

> The siRNA-mediated knockdown experiments referenced again are not a mechanism – showing a single L-R interaction does not show that this is the driver of prognosis. The authors are confusing what interactions may be present (which they suggest through their data) from what interactions are meaningful which is my concern.

3. The numbers are too low to make meaningful conclusions about the relative amount of regulatory T-cells. As the authors likely know, only 10% of leukoplakia lesions progress to cancer, so its unclear what the authors are really trying to support here.

> The authors do not really give a response to this – merely acknowledge this weakness, which is substantial in my view. It is challenging to know if these results are reliable given the low numbers.

4. LGALS7B has been associated with immune suppression and this is not really seen in this study which would be expected. Also, even if able to show this, showing that there is immune exclusion raises a problem of knowing which is first – did the immune exclusion happen before malignancy or after transformation and what's the role of this in tumor progression.

> The authors address this point adequately

5. There is zero genetic (inferred CNV) analysis which is quite surprising as we would expect to see a stepwise progression in this to support the authors methodology.

> The authors do not address this comment. My point is that there should be progressive alterations at the CNV level, with tumor progression. This is well described and not shown or completed.

> The new analysis showing correlations with TP63 and ATP1B3 are not really relevant as these are individual genes not chromosomal changes. siRNA targeting p63 or ATP1B3 is not that informative as these genes have previously been shown to be important in HNSCC, but that does not mean they are drivers of disease which is what is of interest when considering tumor progression.

6. I am hesitant to believe anything about cell proportions because the tumor tissue is likely to be a large piece than the surrounding normal mucosa or leukoplakia (which are both small by nature) and thus subject to distinct effects with sample processing. Unless using a bulk method, I simply cannot believe any proportions of fibroblasts, epithelial cells, or immune cells. This represents a major concern.

> The authors new analysis basically confirms my fear – misrepresentation of cell proportions so any analysis of cell composition is really hard to make much sense of. Yet, this is something used throughout their study. In addition, they miss my key point that heterogeneity may be a more substantial issue when small bx of normal mucosa or leukoplakia are compared to a larger piece of primary tumor.

7. I don't agree that a more epithelial score (2A bottom) shows that cells have acquired a proliferative activity similar to tumors – clinically we know this is not true as these are slow growing lesions, many of which do not transform.

> While I appreciate the authors efforts to address my concern, the changes made highlight my broader view point of lack of rigor in the methodology and approach of this study. These samples were already known to be HPV+ and HPV- yet the authors just grouped these together, and now they note that samples were “confounded by HPV.”

8. It is unclear how many specimens are really paired across all stages and this requires clarification – looks like just a few even though the authors say 23 patients.

> The authors add a table summarizing the pairs, but this further confirms my concerns. There is only one single sample with match leukoplakia and cancer and only 4 with a cancer and matched LN. This is hardly an advance beyond what has been described previously and raises concerns again about the low numbers.

9. CNAs – immune cells should not have CNAs, and LN mets should have CNAs. The fact that this is not seen is very worrisome. Also chr 22 is amplified in the leukoplakia but not in the cancer/LN, again this

would be something that is never described before – namely, losing a chromosome that's been amplified and is very worrisome (Fig 2D).

> Again, while I appreciate the authors efforts – the rigor is lacking. Only after I pointed out that T-cells should not have CNAs, do the authors now remove them. This is basic scRNA-seq bioinformatics and is an error that should not have been made. It raises broad concerns about the approach.

> The authors do not address my comment that chr 22 is amplified in leukoplakia (Fig 2B), yet then somehow lost again in cancer/lymph node. They just rehash their methodology. This is a big problem – as I wrote “this would be something that is never described before – namely, losing a chromosome that's been amplified and is very worrisome” in terms of believing the results.

10. Findings related to TP63 and ATP1B3 are not really validated or supported. There is no data to suggest TP63 can transform HNSCC and showing 2 genes is a straw man approach which lacks rigor.

> The new experiments are somewhat helpful but the link to CNA is not strong as described above.

11. Defining 9 malignant clones is very strange – normally one would define these as specific programs based on NMF or other approaches. This is highly unconventional and confusing, and again raises concerns. Same comments for CAFs and immune cells. In general, HPV positive tumors are thought to have less intratumoral heterogeneity (see abundant work from Jim Rocco and others), again raising concerns.

> To be clear, I understood from the outset that the authors were referring to transcriptional programs not genetic variants. However, it remains strange that these have not been defined in an orthogonal fashion with NMF or other approaches as I suggested. The authors merely say “we used Seurat”. That's not the rigorous answer I was hoping for.

> My point about applying the same approaches for CAFs and immune cells was ignored.

> I find it troubling that for many of my comments, they authors mere say “we removed what you were worried about” – does not feel rigorous.

12. LGALS7B was previously identified by another HNSCC single cell analysis (Bernstein/Regev groups) as part of an epithelial differentiation signature. This was shown to correlate with worse prognosis but was part of a large program. The authors have focused on a single gene with no sense of how this affects biology. Identifying single genes from single cell analyses and using those prognosis or to identify cellular heterogeneity is fraught with error. The same can be said of the studies of CXCL8 as again picking one gene in the program and stating that this is the critical driver is very incomplete and troublesome.

> If these genes are merely markers for the clusters then what are the clusters and what defines them? This is the overriding circular problem of the paper – we do not know what the biological significance of the programs described really is or even what their role in the tumor microenvironment is. Accordingly,

it is hard to know if there is anything biologically meaningful being reported. The two single marker genes certainly don't provide that.

13. EMT in fibroblasts?? I don't believe this has been shown and is a very strange idea. The authors say this "may indicate that EMT in fibroblasts contributes to the malignant conversion of epithelial cells" but fibroblasts are mesenchymal. Just does not make sense, no precedent and no data to support this.

> Again, just very sloppy – the authors state "we mis-interpreted the hallmark gene set analysis result." The idea that CAFs have a number of genes that overlap with EMT and there is a need to decouple these is very very well established. Again, this is NOT EMT, it is merely a mesenchymal feature in fibroblasts. The authors continue to say "EMT expression in fibroblasts may indicate acquisition of CAF like traits" – CAF's do not undergo EMT.

14. The ICC experiments in 4E are strange – I do not see any evidence of colocalization here. There are plenty of CD44 cells that are nowhere near COL1A1 cells and this does not appear to progress with Ca. The tumor is clearly on the left side of these images and I suspect that the CD44 cells in the stroma in the right panel are not tumor cells. Would need to see p63 or cytokeratin to convince me otherwise.

15. Invasion assays have matrigel which is riddled with ECM factors, this makes it a hard system to judge the effects. A better approach would be to co-culture their cell lines and its unclear why this was not done.

> Appreciate these changes, much clearer now.

16. Correlation of COL1A1 and LAIR2 is weak by eye, does not appear carefully done. Is this true in TCGA deconvolution? That would be the real test.

> There are many studies that have deconvolved T cells in TCGA. I don't understand the statement "proportion of deconvolved T cells was very low" given that in bulk RNA-seq data the T cells are well represented. CIBERSORTx used TCGA as have other studies.

17. Details of single cell analysis are severely lacking. The computational approaches are essentially absent.

> The fact that the authors completely missed including these details initially still remains surprising to me, but the new methods are helpful.

Reviewer #2 (Remarks to the Author):

The authors have done significant work to improve the manuscript. They answered all questions and addressed problematic points I commented on last time. Specifically, I appreciate the authors for having clarified the introduction section, which will provide a better understanding of HNSCC etiology to a broader audience and reanalyzed the data following a more consistent approach in both tumor conditions as previously suggested and detail that better in the methods section.

The manuscript is now of a very high quality.

Reviewer #3 (Remarks to the Author):

I appreciate the authors' efforts to address my previous comments. I am generally satisfied, but still required clarification on the points below.

The authors still don't address which T cells are cycling? They have designated a cluster of cycling T cells, but is this CD4? CD8? A mixture? The proliferation genes dominate in clustering to separate them out, but they will still have CD4/8 detectable. It would be relevant to understand if one population is preferentially cycling over others or if each subset generally has a few.

I am not convinced by FACS plots in Fig 6 g and h. 6g looks like compensation between fluorophore channels has been a problem (spillover between channels resulting in the dot plot falling along the $x=y$ line). I would want to see gating strategies for h.

Reviewer #4 (Remarks to the Author): Expert in single-cell RNA-seq analysis

I was asked to comment on the previous exchange between reviewer 1 and the authors. My comments are added to the previous comments in the attached pdf, highlighted in red.

Point by point replies to the reviewers' comments

Reviewer's Comments:

Reviewer #1:

I read "Single-cell transcriptome profiling of the stepwise progression of head and neck cancer" by Choi et al with interest. HNSCC progression is certainly an interesting and important question and I think the authors have a good approach to this problem. Unfortunately, the conclusions are quite vague and rather indeterminate and there are major technical and analytical concerns. The manuscript feels like a mish mash of "one off" observations with no clear story or really validated finding, hampering the novelty and conceptual advance.

I tend to disagree with this general assessment. The general approach used by the authors was to 1) to create an atlas-style dataset of HNSCC; 2) use this dataset for discovery and 3) follow up on interesting lead candidates. In contrast to many single cell atlas style papers from the last years, I was rather content with the level of follow up provided, and clearly, the data driven discovery approach employed here is valuable. Most validations make use of simple cell line models and none of them was developed further into, e.g., *in vivo* models, but this should not be an obstacle for publication of an interesting set of observations in *Nature Communications*. Also, even though the observations regard different aspects of HNSCC (e.g. epithelial cells, immune environment, fibroblasts), the authors nicely tie this together in their final figure 7, and I would not refer to the manuscript as a mish mash.

A weakness of the manuscript is the rather low number of patients. This mostly affects the analyses of malignant cells (figure 3), since a lot of inter-patient variability is clearly observed here, and many cell states will not be covered. I comment on this point further below, in the context of point 6.

1. Association of poor prognosis with a single gene is riddled with errors and issues with false discovery. I am underwhelmed by the idea that *LGALS7B* and *CXCL8* correlate with outcome as it is well established that fibroblasts in general lead to worse outcomes based on the TCGA and other studies.

Response:

We have demonstrated that the proportions of the malignant cell cluster (CC1) and fibroblast cluster (CF1) are associated with HNSCC prognosis. However, we did not suggest that a single gene is predictive for tumor prognosis. *LGALS7B* and *CXCL8* were the most prominently expressed genes in CC1 and CF1, respectively. These genes have been shown to harbor onco-promoting functions,

supporting the aggressive feature of CC1 and CF1 clusters. We validated the aggressive phenotypes of the patients with abundant CC1 or CF1 cells using independent data sets, which supported the robustness of our findings.

I do agree with the reviewer that the survival analyses included in this manuscript are rather weak. The approach used in the revised version, i.e. stratification into CF1-low and CF1-high, is clearly superior to stratification by a single gene. However, before stratifying patients into CF1-low and CF1-high or CC1-low and CC1-high, patients should be stratified into fibroblast low and fibroblast high, and also HPV+ and HPV-. Ideally, a multivariate analysis should be performed.

2. Interaction with CD44+ malignant cells in and of itself if not a mechanism. Furthermore, over the past 2 decades, there has been significant doubt over whether these are truly cancer stem cells given that these properties are not appreciated in patients. For example, do the authors see that such cells are stem cells by their own analysis? I doubt it as no prior HNSCC single cell study has shown that.

Response:

In the receptor-ligand interaction database, *COL1A1* and *CD44* were the putative ligand-receptor pair. However, their functional interaction between them was not studied thoroughly. In this study, we demonstrated the interaction between *COL1A1* and *CD44*, using siRNA-mediated knockdown experiments using different systems such as conditioned media and co-culture experiments. These findings strongly suggest the *COL1A1-CD44* interaction.

Previously, many studies have shown that CD44 is a cancer stem cell marker and is associated with aggressive cancer behaviors. However, we did not investigate cancer stem cell-like characteristics of the CD44 expressing cells. In the Discussion section, we explained that these cells might be related to cancer stem cells. However, we removed it from the revised manuscript to clarify the manuscript.

I do not quite understand the argument of the reviewer, since the authors do not investigate stemness. The analysis regarding *COL1A1* and *CD44* appear sound. A (more detailed) suggestion would be, to include a plot demonstrating that CD44 expression on cancer cells also increases during tumor progression. The authors write that there is “interdependent and stepwise expression” of these factors, but the figures do not show any type of correlation between the *COL1A1* expression on fibroblasts and the *CD44* expression on tumor over tumor progression.

3. The numbers are too low to make meaningful conclusions about the relative amount of regulatory T-cells. As the authors likely know, only 10% of leukoplakia lesions progress to cancer, so its unclear what the authors are really trying to support here.

Response:

We agree with the reviewer's comment that our study has limitations in sample size and the number of cells. Although the sample size was limited, we demonstrated the stepwise pattern of altering T cell repertoires during cancer progression. From the trajectory analysis, we identified that the *LAIR2* expression in Tregs facilitates Treg expansion and tumor progression. Our strategy to identify the stepwise patterns could successfully determine the functional significance of *LAIR2* in Tregs.

We agree that only 10% of the leukoplakia progresses to cancer; thus, the features observed in the leukoplakia lesions may not necessarily be related to the cancer development. However, it is generally agreed that the multiple genetic or environmental aberrations may accumulate during cancer progression, providing tumor-promoting environments. Our and previous studies have suggested that the early events of DNA copy number aberrations with concomitant transcriptional deregulation in the premalignant lesions are likely to play potential driver roles during cancer progression. In addition, we validated the functional significance of our findings by performing cell culture experiments and computational analyses.

I do not see technical weaknesses in the analysis regarding Tregs. The plots in figure 6b show that the presence of Tregs increases already at the LP stage; a p value for the difference between NL and LP should be provided. It is interesting to see changes in the immune environment already at a precancerous stage. Maybe the implications of this finding could be discussed in more detail.

4. *LGALS7B* has been associated with immune suppression and this is not really seen in this study which would be expected. Also, even if able to show this, showing that there is immune exclusion raises a problem of knowing which is first – did the immune exclusion happen before malignancy or after transformation and what's the role of this in tumor progression.

Response:

We appreciate the reviewer's comment. Unfortunately, we did not investigate the immunosuppressive roles of *LGALS7B*. However, we assessed whether the CF1 and CC1 abundances are associated because both CF1 and CC1 abundances had aggressive features. As expected, CC1 and CF1 abundances were significantly correlated in the pooled HNSCC data ($r = 0.41$, $P = 1.11 \times 10^{-38}$, Fig. 5f).

In addition, we demonstrated that galectin-7 treatment significantly induced IL-8 expression in both HNSCC and CAF cells (*i.e.*, CAF30, CAF57, CAF58, and CAF70, **Fig 5g**). These results suggest that galectin-7 activates both malignant cells (CC1) and fibroblasts (CF1) to express IL-8, resulting in aggressive progression of HNSCC. We added this data in the revised manuscript.

The new analysis added by the authors is quite interesting and should address this point. The observation regarding Tregs (see point 3) seems to indicate that changes to the immune environment occur before malignancy.

5. There is zero genetic (inferred CNV) analysis which is quite surprising as we would expect to see a stepwise progression in this to support the authors methodology.

Response:

Previously, genetic DNA copy number alterations in 3p, 3q, and 8p have already been reported, which we could confirm in our data. Following the reviewer's comment, we validated our findings using the HPV-negative TCGA-HNSCC data, showing significant correlations between DNA copy numbers and transcription levels of *TP63* ($r = 0.19$, $P = 2.5 \times 10^{-3}$) and *ATP1B3* ($r = 0.37$, $P = 2.0 \times 10^{-9}$, **Fig. 2f**). We also validated the functions of *TP63* and *ATP1B3* by performing siRNA-mediated knockdown experiments. Treatment with siRNAs targeting *TP63* (si*TP63*) or *ATP1B3* (si*ATP1B3*) suppressed tumor-promoting functions of HNSCC cells, including proliferation, migration, invasion, and sphere formation (**Fig. 2g, h, i**). Thus, we suggest that the CNA-dependent expression of *TP63* and *ATP1B3* plays a critical role in HNSCC progression. We added the data in the revised manuscript.

The new analysis added by the authors is quite interesting and should address this point.

6. I am hesitant to believe anything about cell proportions because the tumor tissue is likely to be a large piece than the surround normal mucosa or leukoplakia (which are both small by nature) and thus subject to distinct effects with sample processing. Unless using a bulk method, I simply cannot believe any proportions of fibroblasts, epithelial cells, or immune cells. This represents a major concern.

Response:

We agree with the reviewer's comments. Cell composition in individual samples can be highly heterogeneous according to the sample preparation and experimental procedures. We prepared the large piece from the tumor tissues, but only a small proportion of the cells were profiled. We agree

that cell proportion obtained from scRNA-seq may not represent the actual proportion in tissues. With this concern, we compared the cell composition using deconvolution analyses of the bulk RNA-Seq data. Indeed, only a small number of malignant cells were detected in the scRNA-seq data. However, deconvolution analysis of bulk RNA-seq showed that malignant cells were the most common (79.17%), followed by fibroblasts (10.07%) (**Supplementary Fig. 4a**). We added this statement in the revised manuscript.

This is indeed a very important point, but the new analysis added by the authors has mostly addressed it. A weakness that remains in the manuscript that the number of patients probably does not allow for a discovery of all cell states of malignant cells. For example, cluster CC2, CC3, CC4 and CC5 are only represented by single patients, so probably there are many more cell states in HNSCC that the authors miss. This limits the deconvolution analyses that make use of these clusters.

One solution to this could be to sort out epithelial cells from more (e.g. 50-100) tumor samples and perform highly multiplexed single cell RNA-seq of 100-500 cells per tumor. To make this a “fast” experiment, samples can for this simply be pooled and sorted/processed together. Following single cell RNA-seq, SNPs can be used for demultiplexing into donor, e.g. using the Vireo tool. **Alternatively, it could also be sufficient here to discuss this issue explicitly.**

7. I don't agree that a more epithelial score (2A bottom) shows that cells have acquired a proliferative activity similar to tumors – clinically we know this is not true as these are slow growing lesions, many of which do not transform.

Response:

We greatly appreciate the reviewer's comment. We found that the data was confounded with HPV infection status; thus, we re-analyzed the epithelial and proliferation scores of the epithelial/malignant cells according to the HPV infection status. We found that the HPV-negative LP tissues had higher epithelial scores than those of NL tissues, implying the transformation of the epithelial cells in LP tissue ($P < 5.03 \times 10^{-4}$, NL vs. LP, **Fig. 2a, left**). However, the proliferation scores of the LP tissues were not much altered compared to those of the NL tissues, which agrees with the slow-growing characteristics of the LP tissues (**Fig. 2a, right**). We added these data and statements in the revised manuscript.

I would recommend to remove the analysis of epithelial scores, it is inconclusive and hard to understand.

8. It is unclear how many specimens are really paired across all stages and this requires clarification – looks like just a few even though the authors say 23 patients.

Response:

We apologize for any inconvenience caused by the insufficient sample information. We added the details on the pair-matched sample information and the patient's clinicopathological information in **Supplementary Table 1**.

9. CNAs – immune cells should not have CNAs, and LN mets should have CNAs. The fact that this is not seen is very worrisome. Also chr 22 is amplified in the leukoplakia but not in the cancer/LN, again this would be something that is never described before – namely, losing a chromosome that's been amplified and is very worrisome (Fig 2D).

Response:

We appreciate the reviewer's valuable comment. The T cells having CNAs (449 of 17,869 cells, 2.51%) were re-assigned as the undetermined cells and removed in the analysis.

Single cell CNV calling tools have certain false positive rates, this is certainly acceptable.

We observed the DNA copy gain at chr22 in CA and LN tissues (C30, C38, C15, and LN38); however, these CNAs were only observed in a few cells (3.92 %). This might be due to the heterogeneous cell proportion across the samples. To overcome this limitation, we modified the CNA calling method to detect the recurrent CNAs, considering the different cell proportions in each sample. We determined the tissue type-specific CNAs and the sample-specific CNAs, respectively. Tissue-specific CNAs were determined by comparing the frequencies of the cells with CNAs in each tissue (*i.e.*, LP vs. NL and CA/LN vs. LP) with a cutoff of the fold difference of the cell proportion with CNAs greater than 10 % and Fisher's exact test ($P < 0.001$) between the tissues. In addition, the sample-specific CNAs in the epithelial cells were determined with a cutoff of a cell proportion with CNA greater than 5 %. Then, we defined the recurrent CNAs as the genes observed in at least two samples with the overlap between the tissue-specific and the sample-specific CNAs. We described this statement in the revised **Supplementary Methods**.

I cannot follow the debate here. Why should a subclone present in the carcinoma in situ stage not disappear in later stages of tumor evolution? At what frequency was this clone observed, might it be simply false positives? On the other hand, I also cannot follow the description the authors provide on

their modified CNA analysis strategy. In a context like this it is advisable to work with established methods.

10. Findings related to TP63 and ATP1B3 are not really validated or supported. There is no data to suggest TP63 can transform HNSCC and showing 2 genes is a straw man approach which lacks rigor.

Response:

Following the reviewer's comment, we validated our findings using the HPV-negative TCGA-HNSCC data, showing significant correlations between DNA copy numbers and transcription levels of *TP63* ($r = 0.19$, $P = 2.5 \times 10^{-3}$) and *ATP1B3* ($r = 0.37$, $P = 2.0 \times 10^{-9}$, **Fig. 2f**). We also validated the functions of *TP63* and *ATP1B3* by performing siRNA-mediated knockdown experiments. Treatment with siRNAs targeting *TP63* (si*TP63*) or *ATP1B3* (si*ATP1B3*) suppressed tumor-promoting functions of HNSCC cells, including proliferation, migration, invasion, and sphere formation (**Fig. 2g, h, i**). Thus, we suggest that the CNA-dependent expression of *TP63* and *ATP1B3* plays a critical role in HNSCC progression. We added the data in the revised manuscript.

I found this data quite interesting at first reading. This could be followed up a bit further, e.g. what do these genes do and why are they overexpressed in CNV? But it is not essential to the manuscript.

11. Defining 9 malignant clones is very strange – normally one would define these as specific programs based on NMF or other approaches. This is highly unconventional and confusing, and again raises concerns. Same comments for CAFs and immune cells. In general, HPV positive tumors are thought to have less intratumoral heterogeneity (see abundant work from Jim Rocco and others), again raising concerns.

Response:

We appreciate the reviewer's valuable comment. We have identified the malignant cell clusters representing the transcriptional clusters, not the genetic clones. Although we have intended to describe the 'intratumoral transcriptional heterogeneity, it is confusing with the previous term 'intratumoral clonal heterogeneity'. Data of genetic variants might be required to evaluate the association of HPV infection with intra-tumoral clonal heterogeneity. Thus, we corrected the term 'malignant clone' to 'malignant clusters' in the revised manuscript. To make the manuscript more concise, we removed the 'intra-tumoral heterogeneity (ITH)' data in the revised manuscript.

We re-analyzed the malignant cell clusters to identify the clusters showing distinct expression of the

marker genes, revealing six malignant cell clusters (CC0 to CC5, **Fig. 3a, top**, see **Supplementary Methods**). We used the cluster algorithms implemented in 'Seurat' package, which is now generally used for the clustering analysis of single cell data.

This is clear now, but see my concern in response to point 6.

12. *LGALS7B* was previously identified by another HNSCC single cell analysis (Bernstein/Regev groups) as part of an epithelial differentiation signature. This was shown to correlate with worse prognosis but was part of a large program. The authors have focused on a single gene with no sense of how this affects biology. Identifying single genes from single cell analyses and using those prognosis or to identify cellular heterogeneity is fraught with error. The same can be said of the studies of *CXCL8* as again picking one gene in the program and stating that this is the critical driver is very incomplete and troublesome.

Response:

We agree with the reviewer's comment. We identified *LGALS7B* and *CXCL8* as expression markers for the abundance of CC1 and CF1, respectively. We demonstrated that the proportions of CC1 and CF1 in tumor tissues were associated with an unfavorable prognosis of HNSCCs. Although *LGALS7B* and *CXCL8* were not identified as prognostic markers, tumor-promoting functions of these genes have been shown previously, supporting our findings. Thus, we suggest that the increased proportion of CC1 and CF1 clusters may contribute to the aggressive progression of HNSCC.

In the version of the manuscript I saw, there was no excessive emphasis on single gene analysis, but rather, CC1 and CF1 abundance were scored. This seems related to the reviewer's first point.

13. EMT in fibroblasts?? I don't believe this has been shown and is a very strange idea. The authors say this "may indicate that EMT in fibroblasts contributes to the malignant conversion of epithelial cells" but fibroblasts are mesenchymal. Just does not make sense, no precedent and no data to support this.

Response:

We greatly appreciate the reviewer's valuable comments. We mis-interpreted the Hallmark gene set analysis result. As we have shown, we observed that EMT gene sets were stepwisely expressed in fibroblasts but not in malignant cells during cancer progression. Previously, the cells undergone EMT

were very similar to cancer-associated fibroblasts (CAF), expressing many mesenchymal or EMT markers (Pastushenko et al. *Trends Cell Biol.* 29, 212–226, 2019). A recent study has also shown that the well-defined EMT signatures coexist with the EMT programs of cancer cells and CAF programs, suggesting the need to decouple the mesenchymal expression profiles of cancer cells and CAFs (Tyler et al. *Nature Comm*, 2592, 2021). Indeed, the EMT-related genes included many CAF-related genes (e.g., *POSTN*, *GREM1*, *STHRC1*, *ADAM12*, and *COL11A1*, **Supplementary Fig. 7, right**); thus, we suggest that EMT expression in fibroblasts may indicate the acquisition of CAF-like traits. We replaced this data and interpretation in the revised manuscript.

I actually also found the part on EMT very confusing and would suggest to exclude it from the manuscript.

14. The ICC experiments in 4E are strange – I do not see any evidence of colocalization here. There are plenty of CD44 cells that are nowhere near COL1A1 cells and this does not appear to progress with Ca. The tumor is clearly on the left side of these images and I suspect that the CD44 cells in the stroma in the right panel are not tumor cells. Would need to see p63 or cytokeratin to convince me otherwise.

Response:

We appreciate for reviewer's comments. We replaced the images with a new one, which showed the expression of CD44 in malignant cells and the enhanced expression of COL1A1 in the nearby located fibroblasts. Following the reviewer's comment, we have demonstrated the localized expression of CD44 and the p63 in the malignant cells (**Supplementary Fig. 8a**).

I lack the expertise to evaluate these data.

15. Invasion assays have matrigel which is riddled with ECM factors, this makes it a hard system to judge the effects. A better approach would be to co-culture their cell lines and its unclear why this was not done.

Response:

Following the reviewer's comment, we evaluated the effects of the siRNAs in the co-culture system of the malignant cells and CAFs. We observed that co-culture of MSKQLL1 and CAF48 increased the tumor cell migration. However, the co-culture of siCD44-transfected MSKQLL1 cells or siCOL1A1-transfected CAF48 cells did not enhance the tumor cell migration (**Fig. 4h**). These results suggest that

the *COL1A1-CD44* interaction between fibroblasts and malignant cells plays a crucial role in HNSCC progression. We added these data and statements in the revised manuscript.

I lack the expertise to evaluate these data.

16. Correlation of *COL1A1* and *LAIR2* is weak by eye, does not appear carefully done. Is this true in TCGA deconvolution? That would be the real test.

Response:

We demonstrated that *COL1A1* expression levels in fibroblasts were significantly correlated with *LAIR2* expression levels ($r = 0.48$, $P = 2.48 \times 10^{-3}$) and the abundance of Tregs ($r = 0.38$, $P = 2.60 \times 10^{-2}$, **Fig. 6e**). In addition, collagen treatment significantly induced *FOXP3* expression in Tregs (**Fig. 6f**). FACS analysis also confirmed that collagen treatment increased the number of Tregs and *LAIR2*+ Tregs (**Fig. 6g, h**). Thus, we suggest that the collagen released from the fibroblasts induces Tregs to express *LAIR2*, providing a favorable microenvironment for tumor progression. We added these data and statements in the revised manuscript.

Unfortunately, we could not perform deconvolution analysis with the T cells in TCGA data, because the proportion of the deconvolved T cells was very low.

This seems OK.

17. Details of single cell analysis are severely lacking. The computational approaches are essentially absent.

Response:

We apologize for lacking detailed information on data analysis. We entirely and carefully re-wrote the details on the computational analysis methods in **Supplementary Methods**. To guarantee the reproducibility of our results, we also submitted the source data used in the manuscript.

The computational methods part now provides sufficient detail.

Reviewer #2 (Remarks to the Author): Expert in single-cell RNA-seq, cancer genomics, and immunogenomics

The manuscript presented by Ji-Hye Choi et al. aims to characterize the stepwise progression of HNSCC by applying single-cell RNA-seq technology in normal, leukoplakia, HNSCC and metastasized HNSCC

samples. The study contributes with new insights into the molecular mechanisms of stromal, immune and malignant cells that are involved in the non-neoplastic lesion to metastatic tumor transition. Therefore, this work broadens the knowledge of HNSCC stepwise progression and makes it possible to exploit these findings from the therapeutic perspective.

1. The abstract and introduction point out the relevance for this research, as well as the motivation behind, and they are clearly structured. However, the HPV infection status and its effect in HNSCC progression is not introduced and well explained, neither the motivation behind including this variable in the study (explained in results).

Response:

We appreciate the reviewer's comments. Indeed, various etiological factors are involved in HNSCC development, including exposure to alcohol or tobacco and human papillomavirus (HPV) infection. In particular, HPV infection has played a critical role in HNSCC progression. HPV-positive patients showed more favorable prognostic outcomes than the HPV-negative patients. Genomic analyses have shown that the innate and acquired antiviral immune responses are suppressed in HPV-positive patients. Considering the mechanistic and clinical impacts of HPV, we carefully analyzed the single cell transcripts according to the HPV infection status. We added this statement in the Introduction section.

Overall the methods are well explained. However, the manuscript lacks clarity in some parts and could be further improved if authors address the following points.

2. Could authors make clear if all the tissues representing the stepwise progression of HNSCC come from the same matched patients?

Response:

We apologize for any inconvenience caused by the insufficient sample information. We added the details on the pair-matched sample information and the patient's clinicopathological information in **Supplementary Table 1**.

3. In the paragraph titled "DNA copy number aberrations (CNAs) during HNSCC progression" (row 105), the authors state they could re-assign cells from CA and LN to malignant cells because of their malignant properties. Can authors clarify which are the malignant properties they are referring to? Are they referring to the proliferative activity of those cells?

Response:

We first assigned the epithelial cells from the whole cell analysis; then, the epithelial cells in CA and LN tissues were re-assigned as the malignant cells. Indeed, the CNAs observed in the epithelial cells may indicate the presence of carcinoma in situ (CIS) cells in LP tissues, which we assigned as CIS cells. We evaluated the CNA scores and proliferation scores, which showed a significant upregulation in CA/LN tissues than NL or LP tissues, indicating their malignant property. To clarify the description, we added this statement in the revised manuscript.

4. How did they infer the CNAs? It should be clearly stated in the main text that these have been inferred (and how) directly from the scRNA-seq data.

Response:

DNA copy numbers were inferred using the “inferCNV” R package, with modifications to remove the chromosomal transcription by gene clusters. We described the details of the modified inferCNV method in the Supplementary Methods because the description was lengthy.

The gene clusters located in a chromosomal region with similar gene functions can be falsely estimated to have DNA copy number aberrations by the inferCNV method. With this concern, we determined the functional gene clusters by calculating the functional associations of the genes located within a chromosomal region of 1 Mb using gene ontology analysis ($P < 0.05$). Before inferring the DNA copy numbers, we removed the genes residing in the putative functional gene clusters. The immune-related RNA biotypes and sex chromosomes were also excluded from the analysis. Then, the gene-level DNA copy numbers in each cell were estimated by the segmented CNA values using the circular binary segmentation algorithm implemented in the “DNACopy” R package. We added this statement in the Supplementary Methods.

5. How did authors obtain clusters in figure 3A? Are these clusters based on their similarity in their transcriptomics profile?

Response:

We sub-classified the epithelial cells of CA and LN tissues, revealing six malignant cell clusters (CC0 to CC5, **Fig. 3a, top**, for details, see **Supplementary Methods**). The clusters were obtained based on the transcriptional similarity of the cells using a shared nearest neighbor (SNN) modularity optimization-

based clustering algorithm. We added this statement in the revised manuscript.

6. For the identification of CA (cancer-associated)-specific upregulated gene sets, the fold difference (CA vs. normal tissue) value is different for HPV- and HPV+ cells. Why are these cutoff values chosen and why are they differently selected depending of the HPV infection presence? [Supplementary file, line 31]

Response:

We appreciate the reviewer's valuable comments. We re-analyzed the data applying the same cutoff values (>0.1) for the HPV-positive and HPV-negative tumors. We replaced the data in the revised manuscript.

7. In the ligand-receptor analysis, to identify the putative-ligand receptor pairs, the fold difference (ligand vs. receptor) in cancer-associated fibroblast and malignant cells (>0.5) is different to the fibroblast (>0.1). Which is the explanation for this? [Supplementary file, line 55]

Response:

We identified putative interactions between malignant cells and fibroblasts based on the expression levels of the ligand-receptor pairs. Data for putative ligand-receptor interactions were obtained from a previous study⁴. First, the average expression levels for each ligand-receptor pair were estimated in the malignant cells and the fibroblasts across the tissue types of NL, LP, and CA/LN, respectively. Then, to determine the interdependent ligand-receptor pairs between CAFs and malignant cells, we filtered the ligand-receptor pairs using the following criteria. (1) The ligands were expressed, but their receptors were not expressed in CAF (ligand vs. receptor, fold difference of average expression > 0.5). (2) The receptors were expressed, but their ligands were not expressed in the malignant cells (receptor vs. ligand, fold difference of average expression > 0.5). (3) The ligands were expressed in LP than NL (LP vs. NL, fold difference of average expression > 0.1). Because the basal expression levels of the ligands and the receptors were highly different according to the tissue types, showing higher expression in tumors tissues (CA and LN) than non-tumor tissues (NL and LP), therefore, we applied different cutoff values to determine the fold differences.

8. For the trajectory analysis, why are 2 different versions of Monocle used in the same analysis for

different steps. Why is not the same version used for the whole analysis? [Supplementary file, line 59 and 66]

Response:

We performed the pseudo-time trajectory analyses for T cells using Monocle v2 (**Fig. 6c**). In **Fig. 4**, we performed a trajectory analysis with the UMAP-based dimension reduction and single-cell state transition analysis by Monocle version 3, because UMAP-based plot was not supported by Monocle version 2 (**Fig. 4a, right**). We could obtain a similar result of pseudo-time trajectory analysis using Monocle version 2 as shown in the following plot. Thus, we suggest that our findings were not specific to the Monocle software versions.

Reviewer #3 (Remarks to the Author): Expert in tumour microenvironment and stroma, cancer-associated fibroblasts, and immune microenvironment

In the manuscript “Single-cell transcriptome profiling of the stepwise progression of head and neck cancer” the authors employed single-cell RNA-seq of normal, leukoplakia, HNSCC, and metastasized HNSCC. They identified alterations in cell composition with disease stage and clones of malignant cells expressing LGALS7B and fibroblasts expressing CXCL8. These were associated with poor outcome. Authors claim that Col1a produced by fibroblasts interacted with CD44+ tumour cells to support progression, and also supported Treg activation and expansion. While the authors describe interesting phenomena at the RNA level, the conclusions overstate the data shown. The manuscript would benefit from a more detailed analysis and discussion of the data, showing differentially expressed genes that define key populations as the disease progresses. Validation could be stronger or at least described more thoroughly in the text.

Specific points:

1. There are 23 patients in total, but it is not clear how many were from LP vs CA and metastasis i.e. is each of the 23 samples from a different patient or are multiple samples i.e. norm adjacent and cancer and met taken from the same patient? Include a table of patient characteristics or at least disease stage for each sample

Response:

We apologize for any inconvenience caused by the insufficient sample information. We added the details on the pair-matched sample information and the patient's clinicopathological information in **Supplementary Table 1**.

2. Supp Fig1b contains important information and would be relevant to put in the main figure 1, expanding on the information it shows.

Response:

We appreciate the reviewer's suggestion. We replaced **Supplementary Fig. 1b** to **Fig. 1c**.

3. There seems to be a significant expansion in the endothelial compartment in LP (Fig 1 C). this is not explored in the manuscript. What could underpin this, and does the RNA seq indicate a significant change in phenotype that accompanies a change in abundance?

Response:

We greatly appreciate the reviewer's comment. The increased proportion of the endothelial cells in LP might be due to the increased angiogenesis during NL to LP transition, as described previously (Thiem et al. *J Oral Pathol Med.*, 46(9), 710–716, 2017). However, when we evaluated the pair-matched samples (N15 vs. LP15), no significant increase in the endothelial cells was observed. Thus, to avoid any possible mis-interpretation of the data, we removed the data **Fig. 1c**, which showed only the average cell composition from the multiple samples in each tissue type. Instead, we replaced them with the data showing the cell composition for each sample (**Supplementary Fig. 1b**).

4. What is the proliferation score of immune cells in the diseased tissue? Answering this would provide further biological insight into the phenomenon of TIL activation and response in situ vs recruitment

Response:

Previously, cycling T cells have been addressed to be highly proliferative, and build up the dysfunctional program and tumor-infiltrating lymphocyte (TIL) activation (Li et al. *cell*, 176, 775-789, 2019). Consistently, we could demonstrate that the expression levels of the cycling T cell markers were significantly correlated with the expression of the dysfunctional T cells markers and the TIL scores,

respectively (**Supplementary Fig. 15**). Although we did not further elaborate on the functions of cycling T cells, our results suggest that the alteration of the T cell repertoires, including cycling T cells, may contribute to the stepwise progression of HNSCC. We added the data in the revised manuscript.

5. Labelling of Fig3A and B could be better or accompanied with more explanation in the legend. I understand there are 8 clone clusters and 3b then shows that LN cluster with the primary tumour, but it is not clear how these are linked to 3A, or how these relate to the patient id; 3A has CC0-8, then 3B has C09, C43 etc. This goes back to the earlier comment of a table detailing patient sample information

Response:

We appreciate the reviewer's comments. **Fig. 3a** was marked by the labels for malignant cell cluster (CC0-CC5), while **Fig. 3b** was marked by the sample IDs (e.g., C09), respectively. We described the details in the Figure legends.

6. What are the top differentially expressed genes between epi with stage or clone? What factors upregulated in the epithelial compartment in norm vs LP – can then infer interactions with immune cells and fibroblasts

Response:

We appreciate the reviewer's valuable comments. We added the data showing the differentially expressed genes (DEGs) for the malignant cell clusters in **Supplementary Fig. 5a**.

We also identified the stepwise expression genes in each cell type during progression from NL to LP and CA tissue. We identified 17 genes (e.g., *KRT6A*) in epithelial/malignant cells and 41 genes (e.g., *POSTN*) in fibroblasts (**Supplementary Fig. 6**). We added these data in the revised manuscript.

7. What are the top differentially expressed genes between HPV+ and – clusters?

Response:

We have stratified the malignant cells into six clusters, which were closely associated with HPV infection status. We identified differentially expressed genes in the malignant cell clusters, revealing the most prominent expression of *CAML1* and *LGALS7B* (galectin-7B) in CC0 and CC1, respectively (Wilcoxon Rank Sum test $P < 0.001$, fold difference > 1 , **Fig. 3g** and **Supplementary Fig. 5a**).

The results suggest that *CAML1* and *LGALS7B* are the top expressed genes for HPV-positive and HPV-negative tumor clusters, respectively. However, we did not further investigate the DEGs and their roles between HPV-positive and HPV-native tumors.

8. I would not say EMT for fibroblasts – they are already mesenchymal. It may be more appropriate to infer activation status.

Response:

We greatly appreciate the reviewer's valuable comments. We mis-interpreted the Hallmark gene set analysis result. As we have shown, we observed that EMT gene sets were stepwisely expressed in fibroblasts but not in malignant cells during cancer progression. Previously, the cells undergone EMT were very similar to cancer-associated fibroblasts (CAF), expressing many mesenchymal or EMT markers (Pastushenko et al. *Trends Cell Biol.*, 29, 212–226, 2019). A recent study has also shown that the well-defined EMT signatures coexist with the EMT programs of cancer cells and CAF programs, suggesting the need to decouple the mesenchymal expression profiles of cancer cells and CAFs (Tyler et al. *Nature Comm.*, 2592, 2021). Indeed, we observed that the EMT-related genes included many CAF-related genes (e.g., *POSTN*, *GREM1*, *STHRC1*, *ADAM12*, and *COL11A1*), indicating the expression of the CAF-like trait in fibroblasts during HNSCC progression (**Supplementary Fig. 7, right**). We replaced this data and interpretation in the revised manuscript.

9. In light of the numerous recent studies that have highlighted significant heterogeneity, the authors should be encouraged to discuss what the differential genes between fibroblast clusters and disease stage are. What are the key gene identifiers of each cluster? These clusters may infer functional specializations.

Response:

We appreciate the reviewer's suggestion. We analyzed the DEGs for the fibroblast clusters. The fibroblasts expressed many CAF programs during malignant conversion, which was shown in **Supplementary Fig. 7**. In addition, we analyzed the heterogeneity of the CAFs. The CAFs were further stratified into five clusters (CF0 to CF4), revealing their representative marker genes (**Fig. 5a** and **Supplementary Fig. 9a**). When we compared the CAF clusters with the previous CAFs identified from Puram's study¹², we found that CF0 and CF4 were identical to the Puram's subtype CAF1 expressing its marker genes (e.g., *CTHRC1*, *COL1A1*, *COL3A1*, *POSTN*, and *MFAP2*) (**Fig. 5b** and **Supplementary Fig.**

9a); however, CF4 was distinguished from CF0, showing its higher proliferation activity (**Supplementary Fig. 9b**). CF2 and CF3 were identical to the Puram's CAF2 and myofibroblast, expressing its marker genes such as *CXCL12* and *NDUFA4L2*, respectively. Notably, we identified a novel CAF subtype, CF1, expressing *CXCL8* (IL-8 coding gene). CF1 showed distinct expression patterns among the CAF clusters, showing a unique branch in the two-dimensional diffusion map embedding analysis³⁰ (**Fig. 5c, left**).

We added these statements in the revised manuscript.

10. Supp Fig 6 is difficult to interpret. It is difficult to determine who has the ligand and who has the receptor. Presenting this data as ligand-receptor pairs as shown in for example doi: 10.1016/j.celrep.2020.107628., or doi: 10.1038/s41467-021-22801-0

Response:

Following the reviewer's suggestion, we added the ligand-receptor information in **Fig. 4c**. Because we sought to identify the interdependent expression of the ligand-receptor pairs between fibroblasts and malignant cells, we want to demonstrate the expression levels of the ligand and receptors in both fibroblasts and malignant cells together. The details on the analysis method used for identifying the interdependent ligand-receptor expression were described in **Supplementary Methods**.

11. Fig.4: CD44 is not co-localized with collagen – this would present as yellow as signals overlap. There are more CD44+ cells within the collagen matrix. As collagen is not a cellular marker, and CD44 can also mark CAFs, additional markers would be needed to confirm if these are tumour or stromal cells to infer anything.

Response:

We appreciate for reviewer's comments. We replaced the images with a new one, which showed the expression of CD44 in malignant cells and the enhanced expression of COL1A1 in the nearby located fibroblasts. Following the reviewer's comment, we have demonstrated the localized expression of CD44 and TP63 in the malignant cells (**Supplementary Fig. 8a**).

12. Migration of CD44 siRNA cells was significantly lower in control conditions. The authors need to

quantify any additional effect imparted by CAFs and collagen.

Response:

We appreciate for reviewer's comments. We added the quantitated results in the revised manuscripts (**Fig. 4h**). The co-culture of MSKQLL1 and CAF48 increased the tumor cell migration; however, the co-culture of si*CD44*-transfected MSKQLL1 cells or si*COL1A1*-transfected CAF48 cells did not. We added these data in the revised manuscript.

13. Fig 5: Are CAF clusters now limited to the tumour samples alone? Does this include LP stages? Or are these solely from published data in Puram. It is not clear to the reader. Again, what are the differentially expressed genes that distinguish each cluster?

Response:

Fig. 5 shows the CAF clusters identified CA and LN tissues not the NL or LP tissues. To clarify the description, we revised the statement as follows. "The CAFs from tumor tissues (CA and LN) were further stratified into five clusters (CF0 to CF4), revealing their representative marker genes (**Fig. 5a** and **Supplementary Fig. 9a**). When we compared the CAF clusters with the previous CAFs identified from Puram's study, we found that CF0 and CF4 were identical to the Puram's subtype CAF1 expressing its marker genes (*e.g.*, *CTHRC1*, *COL1A1*, *COL3A1*, *POSTN*, and *MFAP2*) (**Fig. 5b** and **Supplementary Fig. 9a**); however, CF4 was distinguished from CF0, showing its higher proliferation activity (**Supplementary Fig. 9b**). CF2 and CF3 were identical to the Puram's CAF2 and myofibroblast, expressing its marker genes such as *CXCL12* and *NDUFA4L2*, respectively. Notably, we identified a novel CAF subtype, CF1, expressing *CXCL8* (IL-8 coding gene).

We added these statements and data in the revised manuscript.

14. There is staining for CXCL8 in supp Fig 10a, but without additional markers, the authors cannot conclude what cell population this marks. It looks like immune cells.

Response:

We appreciate the reviewer's comments. Unfortunately, although we have tried to re-examine the expression of IL-8 in fibroblasts, we could not obtain satisfying results. IL-8 is a secretory molecule and only the smaller proportion of the CAFs express the IL-8; therefore, it might not be easy to observe a quantifiable data. Thus, we removed the data in the revised manuscript.

In addition, we demonstrated that the CAFs induce IL-8 expression by treatment of galectin-7 (**Fig. 5g**). From this result, we suggested the interaction of the LGALS7B-expressing malignant cell clusters may activate CAFs. We added this data in the revised manuscript.

15. The conclusion at the bottom of page eight is very strong for the data presented. the authors need to perform more detailed interaction analyses between CC1 and CF1 to try and substantiate such a relationship.

Response:

We appreciate the reviewer's valuable comments. we assessed whether the CF1 and CC1 abundances are associated because both CF1 and CC1 abundances had aggressive features. As expected, CC1 and CF1 abundances were significantly correlated in the pooled HNSCC data ($r = 0.41$, $P = 1.11 \times 10^{-38}$, **Fig. 5f**). In addition, we demonstrated that galectin-7 treatment significantly induced IL-8 expression in both HNSCC and CAF cells (*i.e.*, CAF30, CAF57, CAF58, and CAF70, **Fig 5g**). These results suggest that galectin-7 activates both malignant cells (CC1) and fibroblasts (CF1) to express IL-8, resulting in the aggressive progression of HNSCC.

Following the reviewer's comments, we have performed ligand-receptor interaction analysis between CF1 and CC1. However, we could not validate the identified interactions experimentally. This might be because the analysis was not performed with a sufficient number of cells to obtain reliable results. Therefore, we did not include this data in the manuscript.

16. Which T cell populations are proliferating in Figure 6?

Response:

We revised the **Fig. 6a** as follows. Cells with chromosomal aberrations were re-assigned as the non-determinant cell type and excluded (for details, see **Methods**). Then, the NK/T cell cluster was further stratified into six subclusters; naïve T cells, CD8+CCL5+, CD4+FOXP3+ (regulatory T cells, [Treg]), CD4+CD154+ (follicular helper T cells), cycling T cells, and NK cells (**Fig. 6a, top**, and **Supplementary Fig. 11**). The T cell cluster showing higher proliferation activity was designated as a cycling T cell as previously described³¹ (**Fig. 6a, bottom**). We added this statement in the revised manuscript.

17. The legend for fig 6 needs to more clearly explain what figure panels show (same for other legends); e.g. detail of what was done in panel and statistics. It is not clear whether validation in 6G is at the RNA or protein level. Protein validation should be included and ideally with functional assays to show Tregs are indeed more activated as the authors suggest. How many times was the experiment performed? How were iTregs generated, and purity assessed for each replicate? There is no mention of this in the methods, or what concentration of collagen was used.

Response:

We appreciate the reviewer's valuable comments. Following the reviewer's comment, we added the FACS data, demonstrating LAIR2 protein expression in Tregs. We demonstrated that *COL1A1* expression levels in fibroblasts were significantly correlated with *LAIR2* expression levels ($r = 0.48$, $P = 2.48 \times 10^{-3}$) and the abundance of Tregs ($r = 0.38$, $P = 2.60 \times 10^{-2}$, **Fig. 6e**). In addition, we demonstrated that collagen treatment significantly induced *FOXP3* expression in Tregs (**Fig. 6f**). FACS analysis also confirmed that collagen treatment increased the number of LAIR2-expressing Tregs (**Fig. 6g, h**). We added these data in the revised manuscript.

General comments

- There is a general vagueness in the description and detail of approaches used by the authors.

Response: We appreciate the reviewer's comment. We carefully revised the manuscript thoroughly, adding the details of the analysis methods in the supplementary Methods and Figure legends.

- There is no indication of statistics used or number of times experiments were performed on validation

Response: We added the statistics and the number of replicated experiments in the revised figure legends.

- Figure 5G is not referred to in the legend

Response: We corrected the Figure numbers in the revised manuscript.

- Details of antibody concentrations for staining not included

Response: We added the antibody concentrations in **Supplementary Methods** in the revised manuscript.

- Details of collagen inclusion or concentrations into invasion assay not included

Response: We added the collagen concentration (1 $\mu\text{g}/\text{ml}$) in the revised manuscript.

REVIEWER COMMENTS

Reviewer #1 (Remarks to the Author):

I re-reviewed “Single-cell transcriptome profiling of the stepwise progression of head and neck cancer” by Choi et al. Unfortunately, the conclusions remain very vague, lack novelty, and technical and analytical concerns persist. The authors do not address this major concern.

1. Association of poor prognosis with a single gene is riddled with errors and issues with false discovery. I am underwhelmed by the idea that LGALS7B and CXCL8 correlate with outcome as it is well established that fibroblasts in general lead to worse outcomes based on the TCGA and other studies.

(Reviewer 1) The authors miss the point here – they are using a few genes to define a cluster as aggressive without a clear biological pathway or underpinning to define this cluster. These analyses therefore are not robustly defined, and the validation in another very small dataset is not convincing.

(Reviewer 4) I do agree with the reviewer that the survival analyses included in this manuscript are rather weak. The approach used in the revised version, i.e. stratification into CF1-low and CF1-high, is clearly superior to stratification by a single gene. However, before stratifying patients into CF1-low and CF1-high or CC1-low and CC1-high, patients should be stratified into fibroblast low and fibroblast high, and also HPV+ and HPV-. Ideally, a multivariate analysis should be performed.

Response:

We appreciate the reviewer’s valuable suggestion. Following the reviewer’s comments, we performed multivariate analyses. Because the CC1 is related to HPV infection status, we performed analyses on the HPV- patients. We could demonstrate that CC1 was significantly associated with the prognostic outcomes of the patients (Supplementary Table 2). However, we could not validate the prognostic significance of CF1 in multivariate analyses, therefore, we removed the result for CF1 in the revised manuscript. Survival analyses were performed only to support the aggressiveness of CC1 and CF1, the overall conclusion will not be changed.

2. Interaction with CD44+ malignant cells in and of itself if not a mechanism. Furthermore, over the past 2 decades, there has been significant doubt over whether these are truly cancer stem cells given that these properties are not appreciated in patients. For example, do the authors see that such cells are stem cells by their own analysis? I doubt it as no prior HNSCC single cell study has shown that.

(Reviewer 1) The siRNA-mediated knockdown experiments referenced again are not a mechanism – showing a single L-R interaction does not show that this is the driver of prognosis. The authors are confusing what interactions may be present (which they suggest through their data) from what interactions are meaningful which is my concern.

(Reviewer 4) I do not quite understand the argument of the reviewer, since the authors do not investigate stemness. The analysis regarding COL1A1 and CD44 appear sound. A (more detailed) suggestion would be, to include a plot demonstrating that CD44 expression on cancer cells also increases during tumor progression. The authors write that there is “interdependent and stepwise

expression” of these factors, but the figures do not show any type of correlation between the COL1A1 expression on fibroblasts and the CD44 expression on tumor over tumor progression.

Response:

We appreciate the reviewer’s comment. We aimed to identify interdependent LR pairs between malignant cells and fibroblasts, which exhibit receptor expression in malignant cells, but whose corresponding ligands are expressed only in fibroblasts and not in malignant cells. We could identify the seven LR pairs showing interdependent and stepwise expression of the ligands in fibroblasts (i.e., *COL1A1*, *COL1A2*, *COL6A3*, *THBS1*, *THBS2*, *TNC*, and *LAMA4*) and their corresponding putative receptors in malignant cells (*CD44*, *ITGB1*, *SDC4*, *CD47*, and *ITGA6*, **Fig. 4c**). Remarkably, the *COL1A1-CD44* ligand-receptor pair showed the most prominent and interdependent expression (**Fig. 4d, top**), and which could be validated by Puram's scRNA-seq data (**Fig. 4d, bottom**). Moreover, *COL1A1* expression was most significantly correlated with the pseudo-time of the fibroblasts ($r = 0.57$, $P < 2.2 \times 10^{-16}$, **Supplementary Fig. 9a**). We could successfully demonstrate the correlated expression between *COL1A1* in fibroblasts and *CD44* in epithelial cells ($r = 0.43$, $P = 7.47 \times 10^{-3}$, **Fig. 4e**, and **Supplementary Fig. 9b**). However, *CD44* expression in fibroblasts was not correlated with *COL1A1* expression (**Supplementary Fig. 9c**). These results may suggest that the *CD44-COL1A1* interaction occurs between fibroblasts and malignant cells but not in an autocrine manner. Immunohistochemical analysis also validated the prominent COL1A1 expression in fibroblasts near the CD44-expressing malignant cells (**Fig. 4f** and **Supplementary Fig. 9d**).

3. The numbers are too low to make meaningful conclusions about the relative amount of regulatory T-cells. As the authors likely know, only 10% of leukoplakia lesions progress to cancer, so its unclear what the authors are really trying to support here.

(Reviewer 1) The authors do not really give a response to this – merely acknowledge this weakness, which is substantial in my view. It is challenging to know if these results are reliable given the low numbers.

(Reviewer 4) I do not see technical weaknesses in the analysis regarding Tregs. The plots in figure 6b show that the presence of Tregs increases already at the LP stage; a p value for the difference between NL and LP should be provided. It is interesting to see changes in the immune environment already at precancerous stage. Maybe the implications of this finding could be discussed in more detail.

Response:

We appreciate the reviewers’ comments. We demonstrated that the proportions of cycling T cells and CD4+ T cells were increased. Statistical significance was indicated in **Fig. 6b**.

Although the number of cells is small, our results show the stepwise increase according to the tumor progression. Cell experiments also consistently support our results.

4. LGALS7B has been associated with immune suppression and this is not really seen in this study which would be expected. Also, even if able to show this, showing that there is immune exclusion raises a

problem of knowing which is first – did the immune exclusion happen before malignancy or after transformation and what's the role of this in tumor progression.

(Reviewer 1) The authors address this point adequately

(Reviewer 4) The new analysis added by the authors is quite interesting and should address this point. The observation regarding Tregs (see point 3) seems to indicate that changes to the immune environment occur before malignancy.

Response:

We appreciate the reviewer's comments. We demonstrated stepwise alteration of T cell repertoires, particularly Tregs, during HNSCC progression, implying the changes in the immune environment before malignancy. We addressed this point in the Results and Discussion sections.

5. There is zero genetic (inferred CNV) analysis which is quite surprising as we would expect to see a stepwise progression in this to support the authors methodology.

(Reviewer 1) The authors do not address this comment. My point is that there should be progressive alterations at the CNV level, with tumor progression. This is well described and not shown or completed.

The new analysis showing correlations with TP63 and ATP1B3 are not really relevant as these are individual genes not chromosomal changes. siRNA targeting p63 or ATP1B3 is not that informative as these genes have previous been shown to be important in HNSCC, but that does not mean they are drivers of disease which is what is of interest when considering tumor progression.

(Reviewer 4) The new analysis added by the authors is quite interesting and should address this point.

Response:

We demonstrated that *TP63* and *ATP1B3* at 3q exhibited the most prominent CNA-dependent transcriptional deregulation in the HPV-negative CIS cells. Supporting this, the 3q gain in premalignant lesions has already been reported (Veeramachaneni et. al., Scientific Reports, 2019; Davidson et. al., FEBS J., 284(17), 2017; Weber et. al., Am J Pathol, 153(1), 1998). We could demonstrate the DNA copy alteration and concomitant transcriptional activation of these genes. Our results suggest that the DNA copy alterations of *TP63* and *ATP1B3* play important roles in HNSCC progression.

6. I am hesitant to believe anything about cell proportions because the tumor tissue is likely to be a large piece than the surround normal mucosa or leukoplakia (which are both small by nature) and thus subject to distinct effects with sample processing. Unless using a bulk method, I simply cannot believe any proportions of fibroblasts, epithelial cells, or immune cells. This represents a major concern.

(Reviewer 1) The authors new analysis basically confirms my fear – misrepresentation of cell proportions so any analysis of cell composition is really hard to make much sense of. Yet, this is something used throughout their study. In addition, they miss my key point that heterogeneity may be a more substantial issue when small bx of normal mucosa or leukoplakia are compared to a larger

piece of primary tumor.

(Reviewer 4) This is indeed a very important point, but the new analysis added by the authors has mostly addressed it. A weakness that remains in the manuscript that the number of patients probably does not allow for a discovery of all cell states of malignant cells. For example, cluster CC2, CC3, CC4 and CC5 are only represented by single patients, so probably there are many more cell states in HNSCC that the authors miss. This limits the deconvolution analyses that make use of these clusters.

One solution to this could be to sort out epithelial cells from more (e.g. 50-100) tumor samples and perform highly multiplexed single cell RNA-seq of 100-500 cells per tumor. To make this a “fast” experiment, samples can for this simply be pooled and sorted/processed together. Following single cell RNA-seq, SNPs can be used for demultiplexing into donor, e.g. using the Vireo tool. Alternatively, it could also be sufficient here to discuss this issue explicitly.

Response:

We greatly appreciate the reviewer’s valuable suggestion. We evaluated whether our malignant cell clusters could be found in independent scRNA-Seq datasets. Subtyping of the malignant cell using nearest template prediction (NTP) analysis could re-identify our malignant cell clusters in GSE103322 (77.16%) and GSE164690 (99.10%). This may support that our malignant cell clusters represent most of the malignant cell types in HNSCC (**Supplementary Fig. 5**). We added this statement in the revised manuscript. As suggested by the reviewer, it is greatly helpful to perform the pooled scRNA-Seq profiling to strengthen our data. However, we did not have enough time to conduct that experiments. We felt that our new analysis could sufficiently support our conclusions.

7. I don’t agree that a more epithelial score (2A bottom) shows that cells have acquired a proliferative activity similar to tumors – clinically we know this is not true as these are slow growing lesions, many of which do not transform.

(Reviewer 1) While I appreciate the authors efforts to address my concern, the changes made highlight my broader view point of lack of rigor in the methodology and approach of this study. These samples were already known to be HPV+ and HPV- yet the authors just grouped these together, and now they note that samples were “confounded by HPV.”

(Reviewer 4) I would recommend to remove the analysis of epithelial scores, it is inconclusive and hard to understand.

Response:

We appreciate the reviewers’ comments. Following the reviewer’s comment, we removed the result of epithelial scores in the revised manuscript. It will make the manuscript more concise.

8. It is unclear how many specimens are really paired across all stages and this requires clarification – looks like just a few even though the authors say 23 patients.

(Reviewer 1) The authors add a table summarizing the pairs, but this further confirms my concerns.

There is only one single sample with match leukoplakia and cancer and only 4 with a cancer and matched LN. This is hardly an advance beyond what has been described previously and raises concerns again about the low numbers.

Response:

We agree with the reviewer's point. Our study has limitations with a small sample size and not fully pair-matched samples. Notwithstanding, by performing single cell-level profiling, we could identify the carcinoma *in situ* cells in leukoplakia lesions that were not detected by pathological examination. We also demonstrated the interdependent ligand-receptor interaction of *COL1A1* and *CD44* between fibroblasts and malignant cells, facilitating HNSCC progression. Furthermore, we report that the regulatory T cells in leukoplakia and HNSCC tissues begin to express *LAIR2*, providing a favorable environment for tumor growth. With the support of the experimental validation, we suggest that our results can provide novel pathobiological insights into cell-cell interactions during the stepwise progression of HNSCCs.

9. CNAs – immune cells should not have CNAs, and LN mets should have CNAs. The fact that this is not seen is very worrisome. Also chr 22 is amplified in the leukoplakia but not in the cancer/LN, again this would be something that is never described before – namely, losing a chromosome that's been amplified and is very worrisome (Fig 2D).

(Reviewer 1) Again, while I appreciate the authors efforts – the rigor is lacking. Only after I pointed out that T-cells should not have CNAs, do the authors now remove them. This is basic scRNA-seq bioinformatics and is an error that should not have been made. It raises broad concerns about the approach.

(Reviewer 4) Single cell CNV calling tools have certain false positive rates, this is certainly acceptable.

(Reviewer 1) The authors do not address my comment that chr 22 is amplified in leukoplakia (Fig 2B), yet then somehow lost again in cancer/lymph node. They just rehash their methodology. This is a big problem – as I wrote “this would be something that is never described before – namely, losing a chromosome that's been amplified and is very worrisome” in terms of believing the results.

(Reviewer 4) I cannot follow the debate here. Why should a subclone present in the carcinoma in situ stage not disappear in later stages of tumor evolution? At what frequency was this clone observed, might it be simply false positives? On the other hand, I also cannot follow the description the authors provide on 7 their modified CNA analysis strategy. In a context like this it is advisable to work with established methods.

Response:

We appreciate the reviewers' careful comments. We simply added some filtering options into the conventional inferCNV method, which could remove the falsely called CNAs in non-malignant cells. To clarify the description of the method, we rewrote the method as follows. “The DNA copy numbers

were inferred using an “inferCNV” and a circular binary segmentation algorithm implemented in the “DNACopy” R package. Inferring the copy number variations (CNVs) can falsely estimate the CNAs by the chromosomal gene clusters which have similar gene functions. To overcome this limitation, we filtered out the gene clusters (within 1Mb) in which genes were functionally enriched with a gene ontology term ($P < 0.05$). The genes with immune-related RNA biotypes and the genes that reside in sex chromosomes were also excluded. The DNA copy numbers of the immune cells in NL tissues were taken as the reference, then the CNAs were identified with more than a fold difference cutoff of 0.1 compared to the reference”.

In addition, as reviewer 1 has pointed out, some CNAs observed in LP were not found in LN, which might be due to heterogeneous cell proportions across the samples. Considering the different cell proportions in each sample, recurrent CNAs were determined with a cutoff of 0.05 in at least two samples in each tissue type. We added these statements in the revised manuscript.

10. Findings related to TP63 and ATP1B3 are not really validated or supported. There is no data to suggest TP63 can transform HNSCC and showing 2 genes is a straw man approach which lacks rigor.

(Reviewer 1) The new experiments are somewhat helpful but the link to CNA is not strong as described above.

(Reviewer 4) I found this data quite interesting at first reading. This could be followed up a bit further, e.g. what do these genes do and why are they overexpressed in CNV? But it is not essential to the manuscript.

Response:

We appreciate the reviewers’ comments. We could validate the significant correlations between CNAs and transcription levels of *TP63* ($r = 0.19$, $P = 2.5 \times 10^{-3}$) and *ATP1B3* ($r = 0.37$, $P = 2.0 \times 10^{-9}$) (Fig. 2f). Moreover, by performing siRNA-mediated knockdown experiments, we examined the functions of *TP63* and *ATP1B3* in HNSCC cells. Treatment with siRNAs targeting *TP63* (si*TP63*) or *ATP1B3* (si*ATP1B3*) significantly suppressed tumor-promoting functions of HNSCC cells, including cell viability, tumor sphere formation, migration, and invasion (Fig. 2g, h, i and Supplementary Fig. 3). These results consistently support that the CNA-dependent expression of *TP63* and *ATP1B3* have functional significance in HNSCC progression.

11. Defining 9 malignant clones is very strange – normally one would define these as specific programs based on NMF or other approaches. This is highly unconventional and confusing, and again raises concerns. Same comments for CAFs and immune cells. In general, HPV positive tumors are thought to have less intratumoral heterogeneity (see abundant work from Jim Rocco and others), again raising concerns.

(Reviewer 1) To be clear, I understood from the outset that the authors were referring to transcriptional programs not genetic variants. However, it remains strange that these have not been defined in an orthogonal fashion with NMF or other approaches as I suggested. The authors merely say “we used Seurat”. That’s not the rigorous answer I was hoping for.

My point about applying the same approaches for CAFs and immune cells was ignored.

I find it troubling that for many of my comments, they authors mere say “we removed what you were worried about” – does not feel rigorous.

(Reviewer 4) This is clear now, but see my concern in response to point 6.

Response:

As mentioned in point 6, we demonstrated that the malignant clusters (CC0-CC5) could be found in independent scRNA-seq datasets, which may support the robustness of our findings. We also evaluated the expression of the previous malignant programs identified from Puram’s study, which revealed that the CC0 expressed hypoxia and epithelial differentiation-related genes while the CC1 expressed the partial epithelial-mesenchymal transition (p-EMT)-related genes (Fig. 3f). The p-EMT has been suggested to localize in the leading edge in proximity to CAFs and are promoted through paracrine interactions between CAFs and malignant cells, resulting in aggressive progression of HNSCC.

We added this statement in the revised manuscript.

12. LGALS7B was previously identified by another HNSCC single cell analysis (Bernstein/Regev groups) as part of an epithelial differentiation signature. This was shown to correlate with worse prognosis but was part of a large program. The authors have focused on a single gene with no sense of how this affects biology. Identifying single genes from single cell analyses and using those prognosis or to identify cellular heterogeneity is fraught with error. The same can be said of the studies of CXCL8 as again picking one gene in the program and stating that this is the critical driver is very incomplete and troublesome.

(Reviewer 1) If these genes are merely markers for the clusters then what are the clusters and what defines them? This is the overriding circular problem of the paper – we do not know what the biological significance of the programs described really is or even what their role in the tumor microenvironment is. Accordingly, it is hard to know if there is anything biologically meaningful being reported. The two single marker genes certainly don’t provide that.

(Reviewer 4) In the version of the manuscript I saw, there was no excessive emphasis on single gene analysis, but rather, CC1 and CF1 abundance were scored. This seems related to the reviewer’s first point.

Response:

We appreciate the reviewer’s valuable comments. We aimed to identify stepwise patterns from the scRNA-Seq data. This revealed the alterations of cell type components including malignant and non-malignant cells. Then, by performing subtyping analyses of the malignant cells, we identified the most aggressive CC1 cluster which expressed *LGALS7B* and p-EMT program. We suggest that CC1 plays an important role in HNSCC progression, demonstrating its functional characteristics. CF1 was found as a CAF cluster expressing IL-8. We also demonstrated the crosstalk between CC1 and CF1 by performing cell experiments. These results also support that *LGALS7B* activates CF1 to express IL-8. Although the multifaceted functions of *LGALS7B*, such as differentiation, were not fully evaluated, we suggest that

LGALS7B plays an important role, at least in part, in CC1-CF1 interaction. Considering the context of the cell heterogeneity, our analyses could identify novel mechanisms in the cell-cell interactions that potentially drive cancer progression. Based on the results, we updated Fig. 7, summarizing the cell-cell interactions during HNSCC progression.

13. EMT in fibroblasts?? I don't believe this has been shown and is a very strange idea. The authors say this "may indicate that EMT in fibroblasts contributes to the malignant conversion of epithelial cells" but fibroblasts are mesenchymal. Just does not make sense, no precedent and no data to support this.

(Reviewer 1) Again, just very sloppy – the authors state "we mis-interpreted the hallmark gene set analysis result." The idea that CAFs have a number of genes that overlap with EMT and there is a need to decouple these is very very well established. Again, this is NOT EMT, it is merely a mesenchymal feature in fibroblasts. The authors continue to say "EMT expression in fibroblasts may indicate acquisition of CAF like traits" – CAF's do not undergo EMT.

(Reviewer 4) I actually also found the part on EMT very confusing and would suggest to exclude it from the manuscript.

Response:

We appreciate the reviewers' comments. We removed the EMT data that make the manuscript more concise.

14. The ICC experiments in 4E are strange – I do not see any evidence of colocalization here. There are plenty of CD44 cells that are nowhere near COL1A1 cells and this does not appear to progress with Ca. The tumor is clearly on the left side of these images and I suspect that the CD44 cells in the stroma in the right panel are not tumor cells. Would need to see p63 or cytokeratin to convince me otherwise.

15. Invasion assays have matrigel which is riddled with ECM factors, this makes it a hard system to judge the effects. A better approach would be to co-culture their cell lines and its unclear why this was not done.

(Reviewer 1) Appreciate these changes, much clearer now.

(Reviewer 4) I lack the expertise to evaluate these data.

16. Correlation of COL1A1 and LAIR2 is weak by eye, does not appear carefully done. Is this true in TCGA deconvolution? That would be the real test.

(Reviewer 1) There are many studies that have deconvolved T cells in TCGA. I don't understand the statement "proportion of deconvolved T cells was very low" given that in bulk RNA-seq data the T cells are well represented. CIBERSORTx used TCGA as have other studies.

(Reviewer 4) This seems OK.

Response:

We appreciate the reviewers' comments. In fact, CIBERSORTx analysis has also demonstrated a small proportion of T cells in TCGA data (~ 2%, Cancers 2021, 13, 1230). Thus, the sub-division of the T cells into smaller subgroups of the cell clusters according to the cell states may impede their statistical significance. Thus, we did not perform the subtyping of immune cells using deconvolution analysis.

17. Details of single cell analysis are severely lacking. The computational approaches are essentially absent.

(Reviewer 1) The fact that the authors completely missed including these details initially still remains surprising to me, but the new methods are helpful.

(Reviewer 4) The computational methods part now provides sufficient detail.

Reviewer #2 (Remarks to the Author):

The authors have done significant work to improve the manuscript. They answered all questions and addressed problematic points I commented on last time. Specifically, I appreciate the authors for having clarified the introduction section, which will provide a better understanding of HNSCC etiology to a broader audience and reanalyzed the data following a more consistent approach in both tumor conditions as previously suggested and detail that better in the methods section.

The manuscript is now of a very high quality.

Reviewer #3 (Remarks to the Author):

I appreciate the authors' efforts to address my previous comments. I am generally satisfied, but still required clarification on the points below.

The authors still don't address which T cells are cycling? They have designated a cluster of cycling T cells, but is this CD4? CD8? A mixture? The proliferation genes dominate in clustering to separate them out, but they will still have CD4/8 detectable. It would be relevant to understand if one population is preferentially cycling over others or if each subset generally has a few.

Response:

The T cell cluster showing higher proliferation activity was designated as a cycling T cell as described previously (Li H, et al. *Cell* 176, 775-789, 2019), which showed higher expression of CD8s but lower expression of CD4s. Because the functional characteristics of these cells observed in dysfunctional T cells have been described well in that study, we did not further analyze the cell characteristics.

I am not convinced by FACS plots in Fig 6 g and h. 6g looks like compensation between fluorophore channels has been a problem (spillover between channels resulting in the dot plot falling along the x=y line). I would want to see gating strategies for h.

Response:

We apologize for any inconvenience caused by the insufficient data. We added the figures for gating strategies in **Supplementary Fig. 13**.

Reviewer #4 (Remarks to the Author): Expert in single-cell RNA-seq analysis

I was asked to comment on the previous exchange between reviewer 1 and the authors. My comments are added to the previous comments in the attached pdf, highlighted in red.

I tend to disagree with this general assessment. The general approach used by the authors was to 1) to create an atlas-style dataset of HNSCC; 2) use this dataset for discovery and 3) follow up on interesting lead candidates. In contrast to many single cell atlas style papers from the last years, I was rather content with the level of follow up provided, and clearly, the data driven discovery approach employed here is valuable. Most validations make use of simple cell line models and none of them was developed further into, e.g., in vivo models, but this should not be an obstacle for publication of an interesting set of observations in Nature Communications. Also, even though the observations regard different aspects of HNSCC (e.g. epithelial cells, immune environment, fibroblasts), the authors nicely tie this together in their final figure 7, and I would not refer to the manuscript as a mish mash. A weakness of the manuscript is the rather low number of patients. This mostly affects the analyses of malignant cells (figure 3), since a lot of inter-patient variability is clearly observed here, and many cell states will not be covered. I comment on this point further below, in the context of point 6.

REVIEWERS' COMMENTS

Reviewer #3 (Remarks to the Author):

I do not have the expertise to comment on the change between reviewers 1 and 4, or whether the authors have now adeptly addressed these points.

The authors have now addressed my comments that remained in the previous revision.

Reviewer #4 (Remarks to the Author):

I went through the manuscript by Choi et al. again. I can confirm that it is done according to best practices in the field. The readability and structure of the manuscript has improved compared to the previous version I've seen and my concerns have been mostly addressed, with one exception pointed out below that needs to be discussed.

I realized that in the previous round of review I had not seen the 2nd round of comments by reviewer 1, so my previous comments were only looking at the initial comments of reviewer 1 and the authors' replies. Hence in some cases I just wrote brief comments like "this now seems OK" etc. without commenting again on new concerns raised by Reviewer 1. I apologize for this. I would like to point out the following:

A major concern by Reviewer 1 was the quantifications of cell type proportions. Many of their points are directly or indirectly related to this. They wrote "The authors new analysis basically confirms my fear – misrepresentation of cell proportions so any analysis of cell composition is really hard to make much sense of. Yet, this is something used throughout their study.". In my opinion the authors have done exactly the right thing, which is to use single cell RNA-seq to qualitatively define what cell types exist, and then use bulk deconvolution methods on larger cohorts (CIBERSORTx) for quantitative analysis. This is more accurate than using single cell RNA-seq, because bulk tissue processing is subject to less experimental variability and can follow highly standardized procedures (see also original CIBERSORTx paper). Interesting correlations with survival, gene expression programs etc are then generally performed on the bigger bulk RNA-seq cohort. This strategy was also employed by recent work in other areas of cancer research (e.g. Zeng et al., Nature Medicine 28(6): 1212-1223) and can be considered best practice.

A limitation of this approach that I pointed out is that it requires an exhaustive single cell reference data set containing all cell states observed in the system (my point 6). The new analysis the authors added (Supplementary Figure 5) shows that this limitation remains: Very different clusters from 3rd party single cell RNA-seq datasets all get classified as the same cell state (eg. In GSE103322, 5-6 clusters are labelled as CC1). If the authors' single cell reference data set were exhaustive, each cluster from a 3rd party dataset should match one or several clusters from the reference. The statement "This may support that our malignant cell clusters represent most of the malignant cell types in HNSCC" is a wrong interpretation of these results. These results instead show that their data misses heterogeneity of epithelial cell populations, due to a small number of patients being sampled. The authors do need to state this limitation.

Minor point: A few times they write "significant correlation" when the correlation is really weak ($R \ll 0.7$) but statistically significant. It would be better to in these cases employ the wording "weak but significant correlation"

REVIEWERS' COMMENTS

Reviewer #3 (Remarks to the Author):

I do not have the expertise to comment on the change between reviewers 1 and 4, or whether the authors have now adeptly addressed these points.

The authors have now addressed my comments that remained in the previous revision.

Reviewer #4 (Remarks to the Author):

I went through the manuscript by Choi et al. again. I can confirm that it is done according to best practices in the field. The readability and structure of the manuscript has improved compared to the previous version I've seen and my concerns have been mostly addressed, with one exception pointed out below that needs to be discussed.

I realized that in the previous round of review I had not seen the 2nd round of comments by reviewer 1, so my previous comments were only looking at the initial comments of reviewer 1 and the authors' replies. Hence in some cases I just wrote brief comments like "this now seems OK" etc. without commenting again on new concerns raised by Reviewer 1. I apologize for this. I would like to point out the following:

A major concern by Reviewer 1 was the quantifications of cell type proportions. Many of their points are directly or indirectly related to this. They wrote "The authors new analysis basically confirms my fear – misrepresentation of cell proportions so any analysis of cell composition is really hard to make much sense of. Yet, this is something used throughout their study.". In my opinion the authors have done exactly the right thing, which is to use single cell RNA-seq to qualitatively define what cell types exist, and then use bulk deconvolution methods on larger cohorts (CIBERSORTx) for quantitative analysis. This is more accurate than using single cell RNA-seq, because bulk tissue processing is subject to less experimental variability and can follow highly standardized procedures (see also original CIBERSORTx paper). Interesting correlations with survival, gene expression programs etc are then generally performed on the bigger bulk RNA-seq cohort. This strategy was also employed by recent work in other areas of cancer research (e.g. Zeng et al., Nature Medicine 28(6): 1212-1223) and can be considered best practice.

A limitation of this approach that I pointed out is that it requires an exhaustive single cell reference data set containing all cell states observed in the system (my point 6). The new analysis the authors added (Supplementary Figure 5) shows that this limitation remains: Very different clusters from 3rd party single cell RNA-seq datasets all get classified as the same cell state (eg. In GSE103322, 5-6 clusters are labelled as CC1). If the authors' single cell reference data set were exhaustive, each cluster from a 3rd party dataset should match one or several clusters from the reference. The statement "This may support that our malignant cell clusters represent most of the malignant cell types in HNSCC" is a wrong interpretation of these results. These results instead show that their data misses heterogeneity

of epithelial cell populations, due to a small number of patients being sampled. The authors do need to state this limitation.

Response:

We appreciate the reviewer's valuable suggestion and agree with the point. We added the following statement in the revised manuscript.

Subtyping of the malignant cell using nearest template prediction (NTP) analysis could re-identify our malignant cell clusters in GSE103322 (77.16%) and GSE164690 (99.10%), respectively (**Supplementary Fig. 5**); however, our cell clusters did not cover all the heterogeneity of the malignant cells, which might be due to a small number of patients being sampled.

Minor point: A few times they write "significant correlation" when the correlation is really weak ($R \ll 0.7$) but statistically significant. It would be better to in these cases employ the wording "weak but significant correlation"

Response:

Following the reviewer's comment, we corrected the revised manuscript.